# Multi-omics monitoring of drug response in rheumatoid arthritis in pursuit of molecular remission

Shinya Tasaki[1,5], Katsuya Suzuki[2], Yoshiaki Kassai[3], Masaru Takeshita[2], Atsuko Murota[2], Yasushi Kondo[2], Tatsuya Ando[1], Yusuke Nakayama[1], Yuumi Okuzono[3], Maiko Takiguchi[3], Rina Kurisu[3], Takahiro Miyazaki[3,6], Keiko Yoshimoto[2], Hidekata Yasuoka[2], Kunihiro Yamaoka[2], Rimpei Morita[4], Akihiko Yoshimura[4], Hiroyoshi Toyoshiba[1] & Tsutomu Takeuchi[2]

Sustained clinical remission (CR) without drug treatment has not been achieved in patients with rheumatoid arthritis (RA). This implies a substantial difference between CR and the healthy state, but it has yet to be quantified. We report a longitudinal monitoring of the drug response at multi-omics levels in the peripheral blood of patients with RA. Our data reveal that drug treatments alter the molecular profile closer to that of HCs at the transcriptome, serum proteome, and immunophenotype level. Patient follow-up suggests that the molecular profile after drug treatments is associated with long-term stable CR. In addition, we identify molecular signatures that are resistant to drug treatments. These signatures are associated with RA independently of known disease severity indexes and are largely explained by the imbalance of neutrophils, monocytes, and lymphocytes. This high-dimensional phenotyping provides a quantitative measure of molecular remission and illustrates a multi-omics approach to understanding drug response.

[1] Integrated Technology Research Laboratories, Pharmaceutical Research Division, Takeda Pharmaceutical Company Limited, 2-26-1, Muraoka-higashi, Fujisawa City, Kanagawa 251-8555, Japan. [2] Division of Rheumatology, Department of Internal Medicine, Keio University School of Medicine, 35 Shinanomachi, Shinjuku-ku, Tokyo 160-8582, Japan. [3] Immunology Unit, Pharmaceutical Research Division, Takeda Pharmaceutical Company Limited, 2-26-1, Muraoka-higashi, Fujisawa City, Kanagawa 251-8555, Japan. [4] Department of Microbiology and Immunology, Keio University School of Medicine, 35 Shinanomachi, Shinjuku-ku, Tokyo 160-8582, Japan. [5] Present address: Rush University Medical Center, Rush Alzheimer's Disease Center, Chicago, 60612 IL, USA. [6] Present address: Nektar Therapeutics, San Francisco, 94158 CA, USA. Correspondence and requests for materials should be addressed to H.T. (email: nre05131@nifty.com) or to T.T. (email: tsutake@z5.keio.jp)

Rheumatoid arthritis (RA) is an autoimmune disorder associated with inflamed joints and often accompanied by systemic symptoms[1]. Disease-modifying antirheumatic drugs (DMARDs) have enabled us to reduce disease activity and halt the progression of RA[2–4]; however, unmet needs persist (such as pain, physical functionality, and fatigue), which have not been resolved even with DMARDs[5]. Additionally, sustained remission without drug treatment, drug-free remission, has not yet been accomplished[6], implying that DMARDs treat symptoms of RA but may not fully address molecular mechanisms that characterize patients with RA[7]. Indeed, it is unclear whether the achievement of clinical remission (CR) reflects the state in which molecular profiles are closer to those of healthy individuals (healthy controls; HCs) than to those of RA patients, which we refer to as molecular remission (MR).

There is a substantial lack of understanding regarding the alignment between the clinical and molecular effects of DMARD treatment. Transcriptomics studies have revealed the molecular effects of TNF blockers or tocilizumab (TCZ) in the peripheral blood[8] or synovial tissues[9] of patients with RA. Although these attempts have shown significant alterations in gene expression by drug treatments, because of the lack of data from HCs, the correlation between gene expression levels in patients after drug treatment and those in HCs remains unknown. Thus, knowledge of the molecular aberrations that persist in patients with RA, even after DMARD treatment, is scarce. Additionally, studies of the effects of DMARDs from "omics" perspectives other than transcriptomics are limited[10–12]. Therefore, multi-omics monitoring of changes in molecular features with DMARD treatments as well as sufficient data from HCs is highly in demand to elucidate the molecular foundation for the development of next-generation DMARDs.

Here, we conducted a longitudinal multi-omics study of HCs and patients with RA treated with widely used DMARDs. The DMARDs used in this study include methotrexate (MTX), a first-line DMARD whose target is not clearly understood, and infliximab (IFX) or TCZ, which are biologics targeting tumor necrosis factor receptor and interleukin 6 signaling, respectively. Our integrative analysis revealed a greater effect of IFX and TCZ than MTX on molecular profiles and that molecular profiles after treatment define stable CR. Moreover, we identified molecular signatures that were resistant to DMARDs, as well as their responsible immune cell subsets. This knowledge will facilitate drug discovery and contribute to the development of precision therapy for RA.

## Results

**Molecular characteristics of drug-naive patients with RA.** The overarching goal of our study is to understand the extent to which drug treatments return the molecular phenotypes in RA to the healthy state. To achieve this goal, we first elucidate the molecular features that characterize drug-naive RA based on multi-omics profiling of blood samples from 45 drug-naive patients with RA and 35 HCs (Fig. 1a and Supplementary Table 1). Our measurements encompass entities in three molecular classes: the whole-blood transcriptome (12,486 probes; 45 RA and 35 HC), serum proteome (1070 aptamers; 44 RA and 35 HC), and peripheral cell counts (26 cell types; 34 RA and 35 HC). Absolute cell counts and cell counts relative to white blood cells were used. We identified 6006 transcripts, 255 serum proteins, and 20 cell variables corresponding to 18 cell types that are significantly associated (moderated $t$-test; FDR <0.05) with drug-naive patients with RA compared with HC (Fig. 1b and Supplementary Data 1).

The challenge of utilizing these RA-associated variables to evaluate the therapeutic effect on molecular phenotypes is to quantify the composite effect of drug treatment rather than just the univariate effect. Some drugs might have small effects on RA-associated variables but consistently push them back to the healthy state. In contrast, some drugs might have strong effects, but the directions of those effects are distributed randomly toward the healthy state and the RA state. In this situation, it is difficult to determine which drug has a more beneficial effect on molecular phenotypes. Therefore, a metric that unifies RA-associated variables is required for an objective assessment of the drug effect. To develop such a metric, we constructed statistical models that classify individuals as RA or HC based on a molecular profile by using a partial least-squares regression. Data were split into five subsets; four subsets were used for training, and the remaining subset was used as test data to estimate the accuracy of the model. We used each subset as test data for the model trained by the remaining four subsets and repeated this process three times, resulting in 15 models trained by different subsets of data. The transcript-based, protein-based, and cell-count-based models, respectively, classified subsets into RA or HC with accuracies, on average, of 98.8%, 92.9%, and 86.5% (Fig. 1c). The accuracies of models trained by permuted data were 39.6%, 41.4%, and 51.0%, on average, for the transcript-based, protein-based, and the cell-count-based model, respectively, indicating that our models outperformed random assignment (Supplementary Fig. 1). The prediction accuracies of the 15 models were very similar (Fig. 1c), and the contributions of variables to the predictions were highly correlated (Supplementary Figs. 2–4). This result indicated that the PLSR models captured predictive variables that were generally informative in our data. Therefore, we used the average prediction from the 15 models as a final prediction from each data type. Hereafter, we refer to an ensemble of 15 models as a model. Using each data type as input, our models produced an RA probability ranging from 0 to 1, where an RA probability greater than 0.5 was classified as RA, and a probability less than 0.5 was classified as HC (Supplementary Fig. 5). As we described above, we built the models based on data from 45 patients with RA who had not received any medications, while patients who had been treated with any medication were removed from the training process. In this respect, we examined whether the models introduced upward bias on RA probability for the samples used in the training process. To achieve this goal, we compared RA odds between 45 patients who were used in the training and 22 patients who had been treated with medications but did not respond to them (Supplementary Table 2) and found no significant differences in RA odds estimated by the models (Supplementary Fig. 6). These results indicated that our ensemble models were reasonably accurate and did not introduce significant bias into the training samples.

To understand the function of biomolecules that are important for the classification of RA and HCs, we analyzed the relative contribution of variables in the prediction (Fig. 1d, f, g and Supplementary Data 1). Transcriptional changes are likely to reflect alterations of cell composition in whole blood. To clarify the contribution of each cell type to the levels of the transcripts, we assessed the expression profiles of up- or downregulated genes based on the reference transcriptomes of purified 15 immune cells measured using the Affymetrix HG-U133 Plus 2.0 array (Supplementary Fig. 7). Indeed, transcripts that were upregulated in RA, such as *OSER1-AS1*, *LOC100419583*, and *SUPT20H*, were all highly expressed in neutrophils, and those that were downregulated in RA, such as *EEF2K*, *IRF2BP2*, and *RAI1*, tended to be expressed at low levels in neutrophils and at high levels in natural killer (NK) cells (Supplementary Fig. 8). To confirm this trend globally, the single-sample GSEA method[13] was used to merge the gene

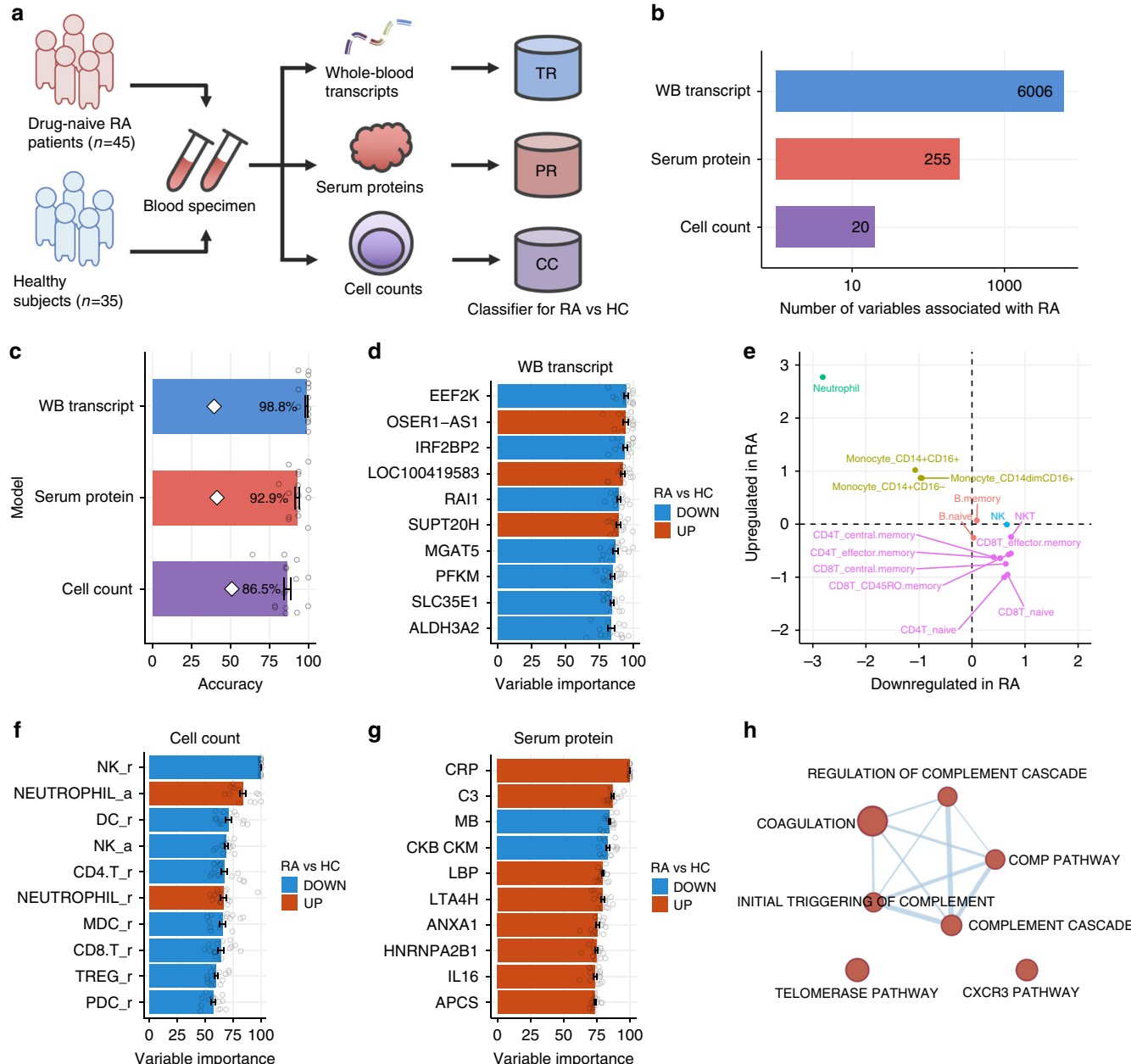

**Fig. 1** Identification of molecular signatures associated with drug-naive patients with RA. **a** Study design. TR, PR, and CC represent the transcript-based model, the protein-based model, and the cell-count-based model, respectively. **b** The number of variables associated with drug-naive patients with RA. A linear regression model was used to compare the levels of variables between RA and HC accounting for age. For the transcripts, the RNA integrity number was also included in the model. The false discovery rate was controlled at 5%. **c** Cross-validation performances of RA diagnostic models. PLSR was employed to build predictive models. Fifteen PLSR models for each data type were generated using 15 different portions of samples as training data. The bar plot represents the average prediction accuracies against the testing data with the standard deviation. White diamonds indicate the expected accuracy of the null models estimated by 1000 sample permutations. **d** The top ten important transcripts for discriminating patients with RA and HC. Error bars represent the variabilities of the contribution to the model prediction that originated from the model ensemble. **e** Expression profiles of important transcripts across 15 immune cells. Meta-expression features of important upregulated or downregulated transcripts in RA were calculated separately using the ssGSEA method and standardized across immune cells. **f** The top ten important cell types for discriminating between patients with RA and HC. A suffix of "r" indicates that the cell counts were normalized to the total number of white blood cells, and a suffix of "a" indicates absolute cell counts. **g** The top ten important serum proteins for discriminating patients with RA and HC. Error bars represent the variabilities of the contribution to the model prediction that originated from the model ensemble. **h** Biological enrichment of influential serum proteins in the model. Serum proteins with a variable importance greater 50 were used for enrichment analysis using hypergeometric test. The biological concepts enriched at the significance level of *p* value <0.05 and FDR <0.05 are displayed. Red nodes represent biological concepts enriched with proteins that are upregulated in RA. Nodes are connected if there are shared genes in two biological pathways. The error bars represent standard errors

expression levels of 2110 genes with an importance measure exceeding 50 for discriminating RA vs HC into a single meta-expression in each of 15 immune cells. The genes that were upregulated in RA were highly expressed and those that were downregulated in RA were weakly expressed in neutrophils (Fig. 1e). These results have also been confirmed at the protein level based on public reference proteomes of 26 immune cells[14] (Supplementary Fig. 9). The cell-count-based model indicated that the decrease in the number of NK cells relative to white blood cells and the increase in the absolute number of neutrophils was associated with patients with RA (Fig. 1f). Together, the cell-count-based model and transcript-based model both indicated an elevation of neutrophils as a hallmark of RA. Indeed, increased absolute numbers of neutrophils, referred as "left shift," are often seen in inflammatory diseases including RA[15]. Known protein biomarkers for the diagnosis of RA, such as C-reactive protein and interleukin 16[16], were identified as highly influential variables in the protein-based model (Fig. 1g). We then investigated the immune cells that contribute to 107 serum proteins with an importance measure greater than 50 based on the reference proteomes[14]. However, both up- and downregulated proteins were highly expressed in the same cell types (Supplementary Fig. 10), suggesting that levels of serum proteins could not be explained simply by cellular protein profiles but are influenced by various tissues such as inflammatory joints or liver[17]. To understand the functions of RA-associated serum proteins, we conducted pathway enrichment analysis using a gene set collection of the canonical pathway and hallmarks from MSigDB[18]. Proteins involved in complement cascade were enriched (hypergeometric test; FDR <0.05) with the upregulated proteins (Fig. 1h and Supplementary Table 3). Activation of a complement system has been reported in patients with RA[19], which indicates that the prediction from the protein-based model reflects the known molecular signature of RA.

**Drug treatments ameliorate the molecular signatures of RA**. To investigate the effects of different treatments on molecular signatures in patients with RA, we collected blood samples longitudinally throughout the course of the MTX, IFX, and TCZ treatments (Fig. 2a and Supplementary Table 4). For each drug, ten patients with RA who were classified as good responders by European League Against Rheumatism response criteria[20] at 24 weeks after the initiation of drug treatment were subjected to multi-omics profiling, and then RA odds were calculated using the models. At 24 weeks of treatment, all three drugs significantly reduced the RA odds estimated based on transcripts, proteins, or immunophenotypes (Fig. 2b). The reduction of RA odds occurred within 4 weeks after drug administration (Fig. 2c). The time course of improvements was similar for the three drugs, but the therapeutic effect of TCZ on serum proteins was significantly greater than those of IFX and MTX throughout the treatments (Welch's t-test; p < 0.05, all weeks). To understand the relationships of treatment effects on the three models, we conducted pairwise comparisons of RA odds. We found a significant consistency between the protein-based model and the cell-count-based model, while the transcript-based model showed modest positive correlations with the other models (Supplementary Fig. 11). Additionally, responders and inadequate responders were significantly separated by the protein-based model and the cell-count-based model (Fig. 2d). Although responders tended to show a greater reduction of RA odds in the transcript-based model than inadequate responders, the difference did not reach statistical significance level. This was not because the drug treatments effectively normalized the transcriptional signatures

associated with RA both in responders and inadequate responders, but the treatment effects on transcriptional signatures were limited even in responders, suggesting the presence of unmet needs at the molecular level in transcriptomes. Taken together, these results indicate that the drug treatments significantly normalize the RA disease signatures of multiple molecular classes, and the magnitude of the therapeutic effect on molecular profiles reflects the clinical response, especially in proteins and cell-counts-based models.

In addition to the model-based assessment of treatment effects, we also characterized the effects of drug treatments at the level of each transcript, protein, and cell type (Fig. 2e and Supplementary Data 2). Approximately 600 transcripts were differentially expressed (FDR <0.05) in patients treated with IFX or TCZ, but no genes exceeded the significance criteria for MTX treatment (Fig. 2e). In TCZ-treated patients, most transcripts were altered in the direction toward the healthy state. Conversely, in IFX-treated patients, a sizable number of those transcripts were altered in the direction away from the healthy state (Fig. 2e). This directional consistency corresponded to a greater reduction in the RA odds in TCZ-treated patients assessed by the transcript-based model (Fig. 2c). Transcriptional changes induced by IFX and TCZ treatments mainly occurred in genes that were expressed at high or low levels in neutrophils (Supplementary Fig. 12a), suggesting that the neutrophil signature was normalized by the drug treatments. The decrease in neutrophil abundance was confirmed by actual cell count data (Fig. 2f), indicating that the drug treatments reduced the neutrophil left shift observed in unmedicated patients with RA (Fig. 1f). TCZ treatment showed a strong effect on serum proteins (Fig. 2e), which was also indicated by the model-based analysis (Fig. 2c). MTX affected a greater number of proteins than IFX, but a sizable number of those proteins were altered in the direction away from the healthy state (Fig. 2e). This directional inconsistency corresponded to a moderate reduction of RA odds in MTX-treated patients, as assessed by the protein-based model (Fig. 2c). Pathway analysis of serum proteins showed that proteins involved in complement pathways were enriched in the proteins affected by IFX and TCZ, but not by MTX (Supplementary Fig. 12b). Complement pathways are also enriched in the proteins associated with unmedicated patients with RA (Fig. 1h), suggesting that IFX and TCZ specifically targeted pathways that were aberrantly activated in RA. IFX and TCZ also affected a greater number of cell types than MTX (Fig. 2e), including neutrophil and NK cells (Fig. 2f), the two most informative cell types in the model (Fig. 1f).

**Relations between molecular remission and disease severity**. Next, we evaluated MR states using RA diagnostic models. Specifically, we defined MR as a state in which the model classifies a patient with RA as an HC (RA probability <0.5). We examined the achievement of MR in good responders and inadequate responders based on their molecular profiles at 24 weeks (Fig. 3a). The three drugs had similar effects based on immunophenotypes, whereas MR at protein and transcriptional levels is achieved only by biologics (Fig. 3b). To clarify the relationships between MR and CR or functional remission, we computed the correlation between MR in each molecular class and CR or functional remission indexes at 24 weeks of treatment. We used disease activity score-28 for RA with the erythrocyte sedimentation rate (DAS28-ESR) for general CR, the clinical disease activity index (CDAI) for CR without considering acute phase response-dependent parameters, and the health assessment questionnaire disability index (HAQ-DI) for functional remission. Molecular

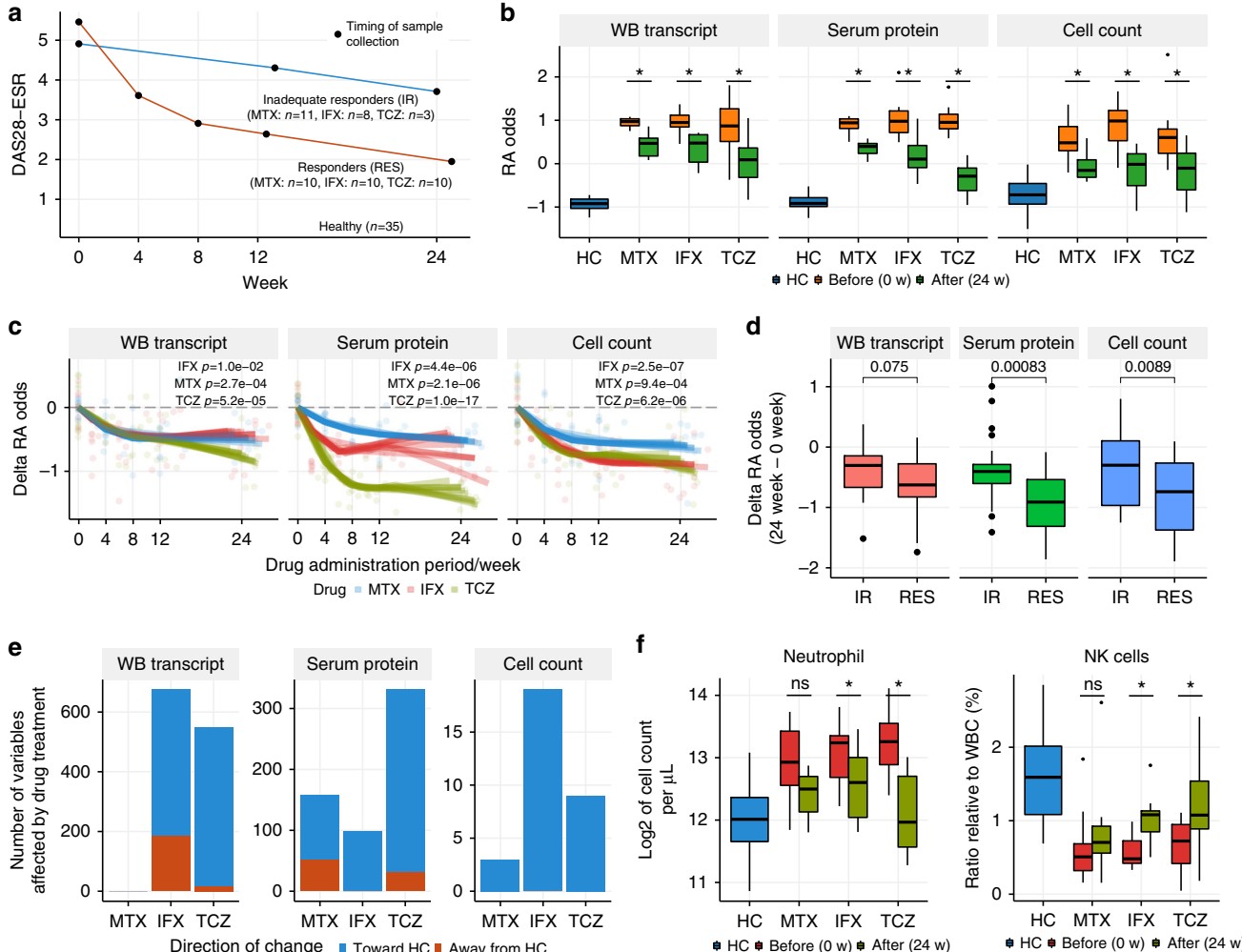

**Fig. 2** Evaluation of the effects of drug treatments on molecular profiles. **a** Sample collection design of the drug response cohort. Responders and inadequate responders to the drug treatments were defined based on EULAR response criteria. Patients who displayed a good response via the criteria at 24 weeks after the first drug administration were classified as responders; others were classified as inadequate responders. The average DAS28-ESR and sampling timing for each group are shown. **b** RA probability changes induced by the treatments. RA probability was transformed to log-odds, and the log-odds at week 0 were compared with those at week 24 using the paired $t$-test (*$p < 0.05$) for each treatment arm ($n = 10$ for each drug). **c** The temporal change in log-odds for being RA during the treatments. The temporal effects of RA odds ($n = 10$ for each drug) were modeled with B-spline smoothing, in which an individual was treated as a random effect. **d** Correlation between the model-based assessments of drug effects and the clinical definition of drug response. The treatment effects on RA odds were compared between responders ($n = 30$) and inadequate responders ($n = 22$) by Welch's $t$-test. **e** The number of variables affected by drug treatments (24 vs 0 weeks). Treatment effect was tested for each drug ($n = 10$) via limma by taking into account the paired samples. RIN value was included in the regression model for testing transcripts. The criterion for significance was set at a $p$ value < 0.05 and FDR < 0.05. **f** Neutrophil and NK cell counts before and after treatment. The asterisk represents a $p$ value < 0.05 and FDR < 0.05. The upper, center, and lower line of the boxplot indicates 75%, 50%, and 25% quantile, respectively. The upper and lower whisker of the boxplot indicates 75% quantile +1.5 * interquartile range (IQR) and 25% quantile −1.5 * IQR

remission defined based on serum proteins were strongly correlated with DAS28-ESR but not with CDAI and HAQ-DI (Fig. 3c). Cell-count-based or transcript-based remission was not associated with CR or functional remission, which suggested that these measures reflected patient characteristics which are not quantified in the clinical indexes. Then, we further examined the individual parameters driving association between protein-based remission and DAS28-ESR. We found that ESR was significantly associated with molecular remission based on the proteins (Supplementary Fig. 13).

To elucidate the benefits of MR, we next addressed whether there were any differences between patients in CR only and those in CR as well as MR. To achieve this goal, we conducted a follow-up study for up to 90 weeks after the final omics profiling (24 weeks), 114 weeks from the initial drug treatment, in the

biologics-treated patients who were in CR based on DAS28-ESR at week 24. These patients had received the same treatments during the follow-up period. At 90 weeks, the patients in MR with multiple molecular classes had a lower DAS28-ESR (linear regression; $p = 0.005$) and HAQ-DI (linear regression; $p = 0.03$) than those who were not in MR (Fig. 3d). To account for the differences in initial disease activity, we included DAS28-ESR, HAQ-DI, or CDAI at the initial time of follow-up in a regression model. In this model, the associations of MR with DAS28-ESR ($p = 0.006$) and HAQ-DI ($p = 0.03$) remained the same or even strengthened for CDAI ($p = 0.03$). The patients who achieved MR in more than two molecular classes exhibited inactive disease states throughout the follow-up period (Fig. 3e). No significant differences were found between IFX and TCZ in DAS28-ESR (Welch's $t$-test; $p = 0.52$) and HAQ-DI (Welch's $t$-test; $p = 0.14$)

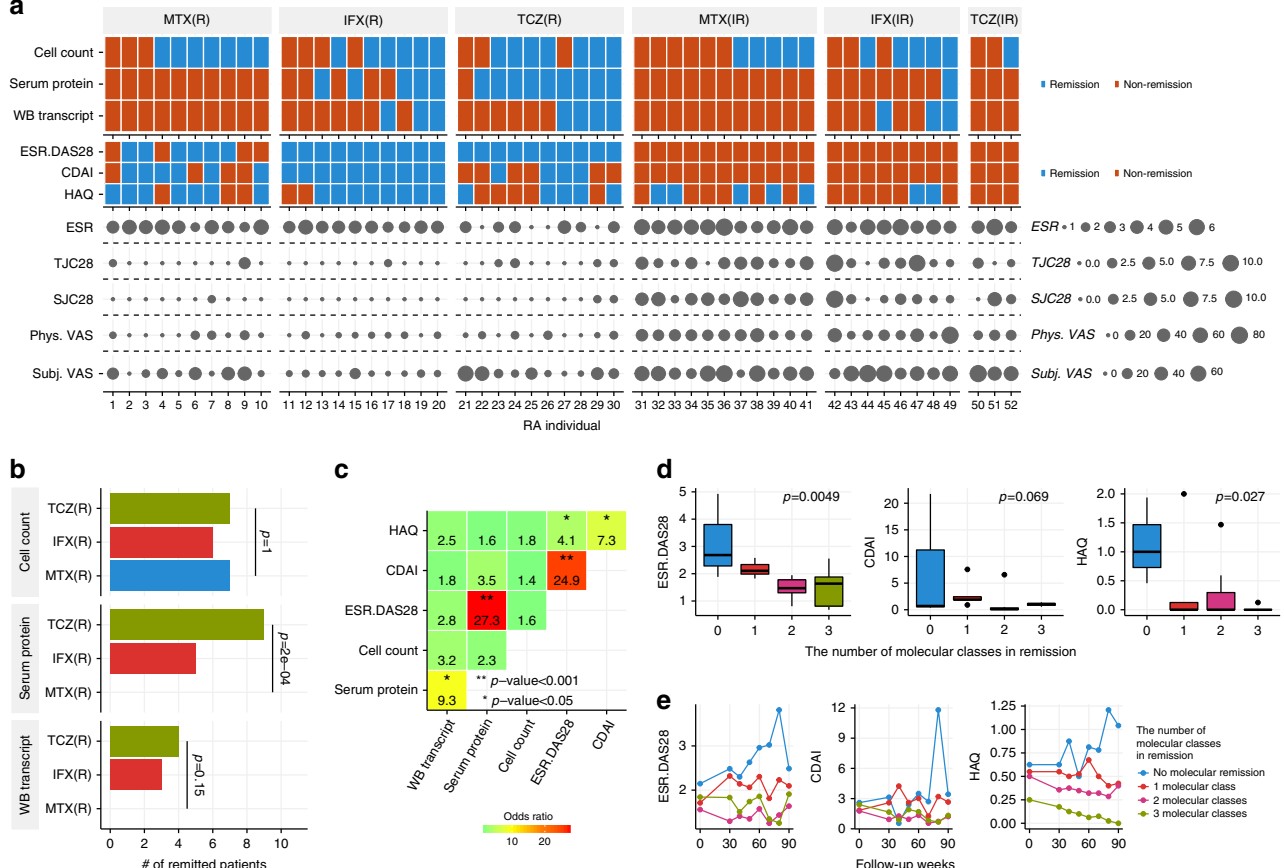

**Fig. 3** Relationship between MR and disease severity indexes. **a** MR profiles along with disease severity indexes at week 24. R and IR indicate drug responders and inadequate responders, respectively. Each column represents MR and disease severity indexes of a single patient with RA. Disease severity indexes include DAS28-ESR, CDAI, HAQ, ESR, tenderness joint counts using 28 joints (TJC28), swollen joint counts using 28 joints (SJC28), physician visual analog scale (Phys. VAS), and subject visual analog scale (Subj. VAS). **b** Comparison of drug effects on the induction of MR. The proportions of patients who achieved remission were compared across three treatment arms by Fisher's exact test. **c** Correlation of remission indexes. The coincidence of remission states based on two different indexes was evaluated by the two-sided Fisher's exact test. The number and color represent the odds ratio, where remissions occurred simultaneously. **d** The MR state is associated with disease activities for patients in CR at 90 weeks. A linear model was used to evaluate the correlation between the number of biological levels that achieved remission states and the disease activities after 90 weeks in RA patients who were treated with biologics ($n = 20$). The upper, center, and lower line of the boxplot indicates 75%, 50%, and 25% quantile, respectively. The upper and lower whisker of the boxplot indicates 75% quantile $+1.5 *$ interquartile range (IQR) and 25% quantile $−1.5 * $ IQR. **e** The temporal changes in DAS28-ESR, CDAI, and HAQ values during the follow-up period, which was split into 10-week intervals. For each patient, the average of multiple measurements of the DAS28-ESR, CDAI, and HAQ values within the same interval was calculated, and then the mean of these values from different individuals was calculated

at 90 weeks, indicating that the correlations between MR and DAS28-ESR and HAQ in the follow-up were not merely due to the differences in long-term drug responses to IFX and TCZ. We also examined the individual parameters of DAS28-ESR reflecting the relationships between long-term CR and MR. The results revealed an association of ESR and tenderness joint counts using 28 joints (TJC28) ($p < 0.05$) with MR status (Supplementary Fig. 14), suggesting that MR not only influenced ESR, but also inflammation status in joints over the long term. These results suggest that MR is associated with long-term stable disease inactivation in patients with RA.

**Identification of residual molecular signatures**. The treatments with biologics significantly normalized the molecular signatures of RA; however, drug treatment did not completely normalize RA odds to the levels seen in HCs (Fig. 2b). To identify these residual RA signatures, we compared the levels of transcripts, proteins, and cell counts in patients with RA at 24 weeks with those in HCs. We found 800 transcripts and 13 serum proteins with

persistent significantly different levels (FDR <0.05) from those in HCs after treatment with MTX, IFX, or TCZ, which we refer to as residual molecular signatures (RMSs) (Fig. 4a and Supplementary Data 3). There were fewer RMSs specific to TCZ treatment than after treatment with IFX or MTX, suggesting that treatment with TCZ could normalize molecular systems that were not affected by IFX or MTX.

Next, we investigated clinical phenotypes associated with RMSs. We calculated meta-features[13] corresponding to the averaged level of transcriptional RMSs and protein RMSs (see Methods) in drug responders at 0 or 24 weeks ($n = 10$ for each drug). The levels of meta-features in RA were significantly different compared with those in HC, and transcriptional RMSs were not significantly normalized by any drug (Fig. 4b). Then, the meta-features were compared with clinical parameters each week. We found that transcriptional RMS and protein RMS showed weak correlations with DAS28-ESR and CDAI at week 0 (Fig. 4c) and other disease indexes (Supplementary Figs. 15 and 16). However, these trends were not observed at week 24. Conversely,

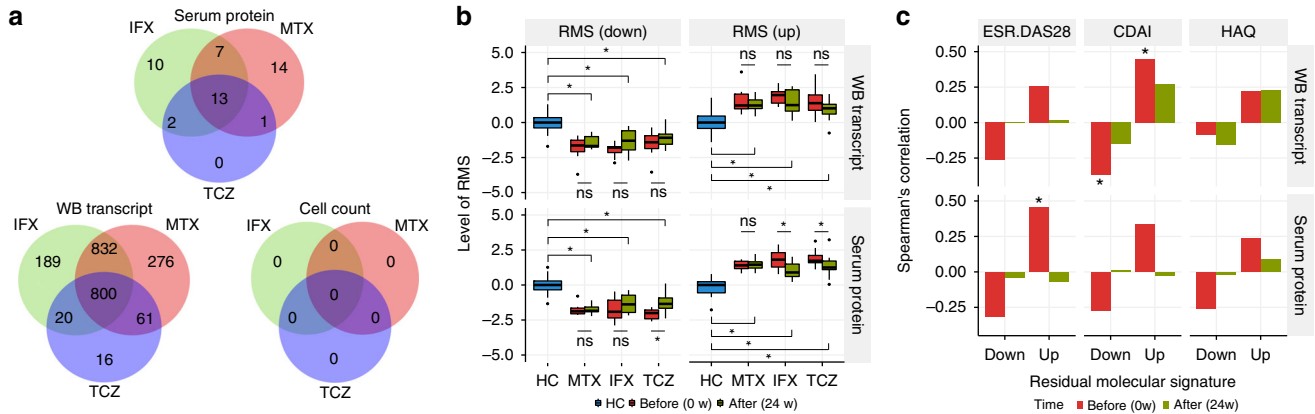

**Fig. 4** Identification of residual molecular signatures. **a** Comparison of residual molecular signatures across treatments. **b** Meta-features of residual molecular signatures were differentially expressed in RA after drug treatment. Meta-features of residual molecular signatures were quantified by summarizing residual transcripts and proteins via the ssGSEA method. The levels of meta-features between RA and HC or before and after treatment were tested by Welch's t-test or the paired t-test, respectively. The asterisk represents a p value <0.05. The upper, center, and lower line of the boxplot indicates 75%, 50%, and 25% quantile, respectively. The upper and lower whisker of the boxplot indicates 75% quantile +1.5 * interquartile range (IQR) and 25% quantile −1.5 * IQR. **c** Remission parameters associated with residual molecular signatures. The levels of meta-features for residual molecular signatures were associated with DAS28-ESR, CDAI, and HAQ-DI before and after treatment (Spearman's correlation; n = 30; *p < 0.05)

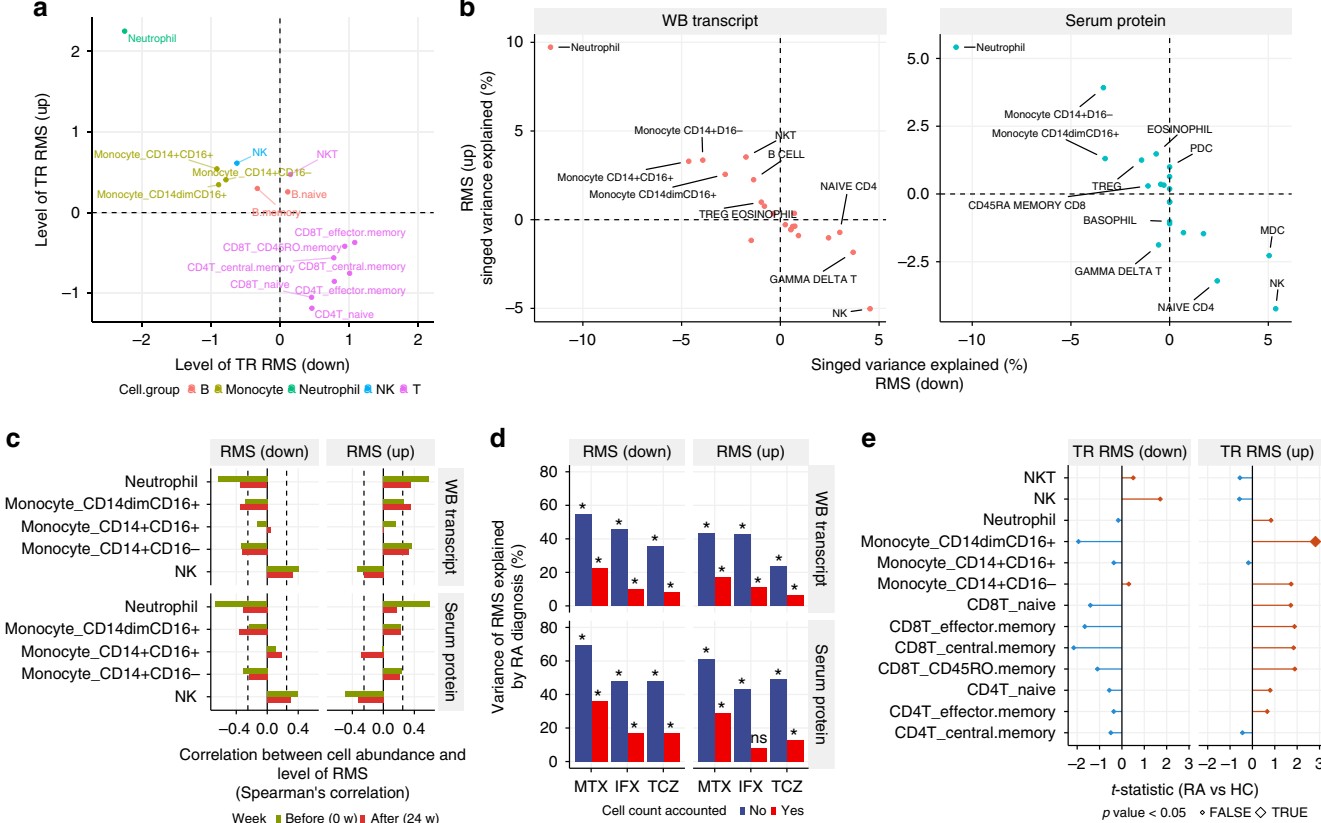

**Fig. 5** Alterations in cell compositions and cellular-level expression explain RMS. **a** Expression profiles of transcriptional RMSs across 15 immune cells. Meta-expression features for the transcriptional RMS that were upregulated or downregulated in RA were calculated separately using the ssGSEA method and standardized across immune cells. **b** The contribution of cell abundance to RMSs. Multivariate linear regression with elastic net regularization was used to estimate the variations in RMSs explained by the absolute cell counts using samples from HCs and drug responders (n = 182). **c** Correlation between RMSs and cell counts of neutrophil, monocytes, and NK cells. **d** The variance in RMS explained by RA diagnosis. The proportion of variance in RMS explained by RA diagnosis was calculated with or without accounting for the contribution of cell counts to RMS. The contribution of cell counts to RMS was accounted for by a multivariate linear regression. The asterisk represents a p value <0.05. **e** Comparison of transcriptional RMSs between patients with RA and HCs in purified immune cells. The levels of transcriptional RMSs in each immune cell were quantified using ssGSEA and compared between RA patients and HCs via a linear model for each cell type

morning stiffness was associated with transcriptional RMS at week 24, but this association was not present at week 0 (Supplementary Fig. 15). Together, these results indicated that RMSs were molecular characteristics of patients with RA that were not associated with specific known disease severities but could not be normalized completely with current symptomatic treatments.

**Cellular alterations explain residual molecular signatures**. We next investigated the cell types associated with RMS using reference transcriptomes and proteomes[14] from purified immune cells. Transcriptional RMS upregulated in RA was highly expressed in neutrophils and monocytes, and transcriptional RMS that was downregulated in RA was weakly expressed in these cell types (Fig. 5a). The same trend was also seen at the protein level (Supplementary Fig. 17). In contrast, protein RMS did not show specific expression in particular cell types (Supplementary Fig. 18). We then examined whether cell counts could explain the expression levels of transcriptional and protein RMSs by using a regularized multivariate regression model (see Methods). Cell counts were estimated to explain 40% and 48% of the variation of up- and downregulated transcriptional RMSs and 33% and 37% of the variation of up- and downregulated protein RMSs, respectively. These fractions are significantly higher than random expectations (permutation $p$ value <0.001). The increase in neutrophils and monocytes and the decrease in NK cells largely explained the levels of transcriptional and protein RMSs (Fig. 5b). Considering both expression specificity and associations with cell counts, the increases in neutrophil and monocytes counts would be the major driver for transcriptional RMS. Indeed, we found significant correlations between transcriptional RMS and neutrophil or monocyte counts at week 0 (Fig. 5c). However, at week 24, correlations between transcriptional RMS and neutrophil counts were weaker than those of week 0, while the relationship with monocytes remained the same (Fig. 5c). This finding suggests that transcriptional RMS after drug treatments is not only due to the left shift in neutrophil, but also the left shift in monocytes, as recently demonstrated in RA[21].

Next, we asked whether cell composition variabilities could explain the observed differences in the levels of RMSs between RA and HC. To evaluate this possibility, we removed cell count effects from RMSs and contrasted the residuals between RA and HC. Given the cell counts, the variabilities in RMSs explained by RA diagnosis decreased but remained significantly high ($p < 0.05$) (Fig. 5d), raising the possibility that expression changes at cellular levels might also contribute to RMS. To test this option, we compared the expression profiles of purified immune cells from RA and HC (Supplementary Table 5) and calculated meta-features for the expression levels of transcriptional RMSs. Although transcriptional changes in each cell type were small, transcriptional RMS tended to be differentially expressed in a direction that was concordant with whole blood in the range of cell subsets tested (Fig. 5e). To investigate a fixed effect shared across cell types, we used a mixed-effect model and found that the fixed effect for the transcriptional RMS upregulated in RA was significant ($p = 0.008$) but that for the one that was downregulated in RA was not ($p = 0.23$). Together, these results indicate that cellular-level expression changes contribute to a fraction of the transcriptional RMS in RA, in addition to the major effect from cell composition alterations.

**Disease-wide landscape of the transcriptional RMS in RA**. Finally, we examined whether the transcriptional RMS observed herein was exhibited by patients with other diseases because molecular signatures shared across diseases often implicate common molecular mechanisms, which would further increase the clinical value of targeting transcriptional RMS. First, we screened the blood transcriptomes of 45 disease conditions using the NextBio database (Supplementary Fig. 19) and identified inflammatory bowel disease (IBD) and obesity as conditions in which transcriptional RMS was increased compared with the controls; uremia was a condition in which transcriptional RMS was decreased compared with the controls (Fig. 6a and Supplementary Data 4). Further investigations revealed that the fold changes in expression of the 800 genes in transcriptional RMS in samples from patients with IBD or obesity relative to their controls considerably resembled those between patients with RA and

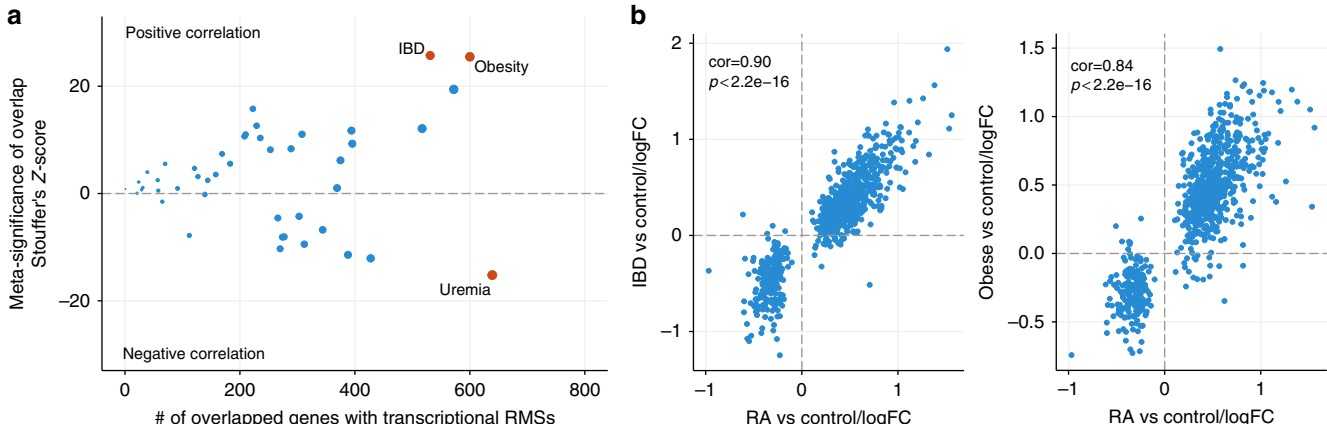

**Fig. 6** The presence of the RA-associated RMS in other disease conditions. **a** The disease-wide landscape of RA RMS. The comparisons of the RA untreatable transcript signature and publicly available disease signatures from whole blood or PBMCs were assessed by Fisher's exact test in the NextBio database. The $p$ values from multiple studies of the same diseases were combined by Stouffer's $z$-score method. The size of the dots is proportional to the number of overlapped genes with the RA-associated untreatable transcript signatures. The red dot represents diseases with $z$-scores that were higher than the top 5% of $z$-scores over all diseases examined. **b** The changes in expression were similar for the transcriptional RMS between RA and the most associated diseases. The fold changes in the 800 transcriptional RMS between patients and controls were calculated using the raw data from the IBD study (GSE33943; $n = 45$ for IBD and $n = 15$ for controls) and the obesity study (GSE18897; $n = 29$ for obese and $n = 20$ for controls). The fold changes from each study were then compared with those determined for our RA cohort ($n = 45$ for RA and $n = 35$ for HCs)

HCs than those in patients with uremia (Fig. 6b and Supplementary Fig. 20). This commonality of molecular signatures suggests that understanding molecular mechanisms underlying transcriptional RMS might be helpful in the treatment of other disorders in addition to RA.

## Discussion

Here, we report the multi-omics analyses of cohorts of untreated and treated patients with RA and HCs. Our longitudinal omics data elucidate the effects of drug treatments at the molecular level. Additionally, an integrative analysis that combines data from different molecular classes and detailed clinical parameters has led to a deep understanding of molecular and cellular systems associated with drug treatment and disease severity. Our approach of profound molecular phenotyping for well-characterized patients is complementary to a large-scale clinical study, which is often achieved at the expense of limited phenotyping and permissive inclusion criteria. Therefore, the relationships between molecules and phenotypes given drug perturbations generated from our multi-omics cohort will assist in the implementation of successful future clinical studies or re-interpretation of existing results.

Deep molecular phenotyping upon drug treatment revealed that TCZ and IFX normalized the molecular profiles in patients with RA more efficiently than MTX at transcriptome, protein, and cell levels (Fig. 2c, e). Furthermore, TCZ could normalize molecular signatures that could not be affected by MTX and IFX (Fig. 4a), suggesting that TCZ is a more potent treatment for RA at molecular levels than the other drugs. We should note, however, that since our molecular measurements were based on peripheral blood, the extent to which the associated molecular signatures reflect those in inflamed joints is unclear. Indeed, TCZ did not induce CR defined by CDAI, a disease index that focuses on joint status, and functional remission defined by HAQ-DI to the same extent as IFX (Fig. 3a). Therefore, further investigations are required to determine whether TCZ can also effectively normalize the molecular signatures in the inflamed joints of patients with RA compared with MTX and IFX.

Our protein-based model shows consistency between the previously developed biomarkers for predicting disease activity in RA[10] regarding the proteins used. Specifically, among the 12 serum proteins used in a model from Bakker et al., ten were measured in our protein panel, and seven of them were within the top 25% influential proteins in the model (Supplementary Data 1). Additionally, both our serum-protein-based model and Bakker's model showed a good correlation with CR (Fig. 3c). The concordance with previous results supports the reliability of our approach.

Multi-layer molecular phenotyping also revealed greater varieties in the types and levels of MR in patients with CR (Fig. 3a). In a prospective follow-up study, we found that the number of molecular classes in MR was associated with stable lower-disease activity over 90 weeks of follow-up (Fig. 3e). As we described above, Bakker's model and our protein-based model use the same proteins as predictors. Interestingly, the risk score from Bakker's model can predict radiographic progression over 2 years of follow-up[22], supporting the potential of peripheral molecular profiles as a predictor of disease progression. However, we should note that the sample size of our follow-up study was limited ($n = 20$) and the number of patients in each subgroup was small; for instance, among the patients treated with TCZ, three patients achieved MR in all three molecular assessments, and only one was without MR. Thus, it is difficult to investigate relationships between MR and clinical outcomes in each treatment and interaction effects of three MR indexes with outcomes. Therefore, a large-scale study with sufficient statistical power is necessary to validate this finding and to evaluate benefit of MR indexes for practical implementation in clinical settings.

We identified treatment-resistant signatures in transcripts and serum proteins (Fig. 4a), and significant fractions of the transcriptional and protein RMSs were explained by cell counts (Fig. 5b). This result indicates that transcriptional and protein RMSs are not caused by alteration of the single cell type but by composite effects of multiple cell types, especially the left shift in neutrophils and monocytes. Transcriptional RMS observed in RA was also observed in IBD and obesity (Fig. 6b). Interestingly, the transcriptomes from patients with IBD originated from the patients in CR[23], suggesting that transcriptional RMS in RA was also not completely normalized in IBD. Because transcriptional RMS is mainly caused by an altered cell composition, especially neutrophils and monocytes, the presence of transcriptional RMS suggests that neutrophil or monocyte counts are also increased in these conditions. Indeed, body mass index shows a positive correlation with neutrophil counts[24,25], and increases in both neutrophil and monocyte counts have been reported in IBD[26]. Interestingly, there are relationships among RA, IBD, and obesity in terms of their development and symptoms. For instance, obesity has been reported to be a risk factor for the development of RA[27]. Additionally, obesity reduces the response to IFX[28] and hinders the achievement of CR[29]. Obesity is also closely associated with the pathogenesis of IBD with respect to comorbidity and responses to drugs[30]. These results suggest that the clinical importance of transcriptional RMS is not limited to RA.

In summary, this study has revealed how the effects of MTX, IFX, or TCZ on three biological levels—transcripts, proteins, and immunophenotypes—are aligned with alterations of disease activities in patients with RA. This knowledge will facilitate the identification of biomarkers for precision therapies for these patients. Our results also clarify the molecular signatures of unmet needs in RA and will contribute to the development of innovative medicines toward the achievement of a deeper MR.

## Methods

**Cohorts**. Sixty-eight patients with RA and 42 HCs who did not have autoimmune diseases or were not receiving any drugs were enrolled from March 2012 to June 2014 (Supplementary Data 5). Of these individuals, 67 patients with RA and 35 HCs were used for analysis of the whole-blood transcriptome, serum proteome, and immunophenotyping, and 45 drug-naive patients with RA who were not being treated with moderate-to-high doses of corticosteroids, immunosuppressants, or biological agents were used for the training models. Of 68 patients, 49 were treated with MTX, IFX, or TCZ, and the whole-blood transcriptome, serum proteome, and immunophenotypes were measured at multiple time points. Of note, two patients were both inadequate responders to IFX and responders to TCZ, and one patient was an inadequate responder to both IFX and TCZ. Of 68 patients with RA and 42 HCs, 14 patients with RA and 16 HCs were used for the transcriptome analysis of immune cell subsets. All procedures were approved by the medical ethics committee of Keio University Hospital and followed the tenets of the Declaration of Helsinki. All samples and information were collected after the patients and HCs provided written informed consent. No statistical methods were used to predetermine sample size.

**Disease phenotyping**. In patients with RA, serum levels of rheumatoid factor (RF) and anti-citrullinated protein antibody (ACPA) were measured before drug treatment. Follow-up evaluations included tenderness and swollen joint counts using 28 joints (TJC28 and SJC28 values, respectively), CRP levels (mg/dl), erythrocyte sedimentation rates (ESRs; mm/h), matrix metalloproteinase 3 (MMP-3) levels (ng/ml), the 28-joint disease activity score (DAS28) with inclusion of the CRP (DAS28-CRP) value or ESR value (DAS28-ESR)[31], the simplified disease activity index (SDAI)[32], the clinical disease activity index[33], and the health assessment questionnaire disability index (HAQ-DI)[34].

**Transcriptome measurements**. For whole-blood transcriptome analysis, blood samples collected from healthy human donors and individuals with RA in PAXgene tubes were frozen and stored at −80 °C. Total RNA was isolated using the PAXgene Blood miRNA Kit (763134, Qiagen, Valencia, CA, USA). Globin-

encoding transcripts were removed from the total RNA by using the Ambion GLOBINclear kit (AM1980, Ambion, Austin TX, USA).

For transcriptome profiling in immune subsets, human whole blood was collected using heparin blood collection tubes (TERUMO, Shibuya, Tokyo, Japan), and immune cell subsets were purified as follows. Peripheral CD8 and CD4 T cells were purified with the CD8 T Cell Isolation Kit and the CD4 T Cell Isolation Kit (130-096-533 and 130-096-495, Miltenyi Biotec, Bergisch Gladbach, Germany), respectively. Monocyte subsets were isolated according to Cros et al.[35]. B cells were magnetically isolated using human CD19 microbeads (130-050-301, Miltenyi Biotec, Bergisch Gladbach, Germany). After staining with Brilliant Violet 421 (BV421)-conjugated anti-CD19 (clone HIB19, BioLegend), allophycocyanin (APC)- and Cyanine 7 (Cy7)-conjugated anti-CD27 (clone O323, BioLegend), fluorescein isothiocyanate (FITC)-conjugated anti-CD38 (clone HIT2, BioLegend) and phycoerythrin (PE)-conjugated anti-human-IgD (clone IA6-2, BioLegend), each subset was isolated with a FACSAriaIII instrument. CD56$^+$CD3$^-$ and CD56$^+$CD3$^+$ NK cells were magnetically isolated using human CD56 microbeads (130-050-401, Miltenyi Biotec, Bergisch Gladbach, Germany). After staining with APC–Cy7–anti-CD3 (clone UCHT1, BioLegend) and PE–anti-CD56 (clone HCD56, BioLegend), each subset was isolated using a FACSAriaIII instrument. For neutrophil isolation, pellets were collected after Ficoll gradient centrifugation and then treated with red blood cell lysis solution (130-094-183, Miltenyi Biotec, Bergisch Gladbach, Germany). Total RNA from the sorted cells was extracted with the miRCURY RNA Isolation Kit (300110, Exiqon, Vedbaek, Denmark), purified with the RNeasy MinElute Cleanup Kit (74204, Qiagen, Hilden, Germany) and amplified with the Ovation Pico WTA System V2™ (3302-A01-NUG, NuGEN Technologies, San Carlos, CA, USA).

The RNA samples were then run on an Agilent 2100 BioAnalyzer using the RNA NanoChip (Agilent, Palo Alto, CA, USA). We further confirmed that the RNA integrity numbers (RINs) were all >7.0. All the RNA samples were hybridized to the Affymetrix human genome U133 plus 2.0 arrays (Affymetrix, Santa Clara, CA, USA).

**Transcriptome data preparation**. Expression values for the probe sets in the Affymetrix human genome U133 plus 2.0 arrays were estimated using frozen robust multiarray analysis (RMA), and their presence call was calculated with the MAS5 algorithm. The systematic bias from experimental batches was normalized for the whole-blood transcriptome data as described below, and stepwise quality control for the probes was conducted as follows. First, the probes that did not match any genes or that targeted multiple genes were removed. The probes that were considered to be absent in more than one-third of samples from HCs and patients with RA were then filtered out. In the case in which multiple probes hybridized to the same gene and were positively correlated with a Pearson's correlation coefficient of more than 0.3, the probe that showed the maximum average signals across the samples was used. Finally, less variable probes with interquartile ranges in the bottom 20% of all probes were filtered out. After application of these quality-control steps, 12,486 probes (corresponding to 10,527 genes) remained. The same procedure was applied to the transcriptomes of the immune cell subsets. The numbers of remaining probes were as follows: 14,964 probes (12,468 genes) from central memory CD4 T cells, 14,632 probes (10,850 genes) from effector memory CD4 T cells, 14,272 probes (10,129 genes) from naive CD4 T cells, 15,403 probes (10,736 genes) from CD45RO$^-$ memory CD8 T cells, 15,403 probes (10,736 genes) from central memory CD8 T cells, 15,144 probes (10,876 genes) from effector memory CD8 T cells, 14,763 probes (10,750 genes) from naive CD8 T cells, 13,271 probes (10,844 genes) from CD14$^+$CD16$^-$ monocytes, 12,958 probes (10,896 genes) from CD14$^+$CD16$^+$ monocytes, 13,083 probes (10,953 genes) from CD14$^{dim}$CD16$^+$ monocytes, 15,923 probes (11,999 genes) from NK cells, 14,938 probes (11,632 genes) from NKT cells and 12,268 probes (9914 genes) from neutrophils.

**Batch effect normalization**. We performed transcriptome and proteome experiments using two separate experimental batches. To estimate the batch effect, the identical RNA or protein samples were included in both batches ($n = 8$ HCs and $n = 12$ patients with RA for the transcriptome analysis; $n = 8$ HCs and $n = 11$ patients with RA for the proteome analysis). Based on the replicated samples, the batch effects were removed from the transcriptome and proteome data using the ComBat method[36]. When replicates were available, the data from the second batch were used for further analyses.

**Serum proteome**. Serum protein concentrations were measured using a slow-off-rate-modified DNA aptamer (SOMAmer)-based capture array (SOMAscan; SomaLogic, Inc., Boulder, CO, USA)[16]. The level of relative fluorescence units (RFUs) that corresponded to a serum protein concentration of 1100 was log$_2$-transformed and used for the analysis[37].

**Immunophenotyping**. Human whole blood was collected using heparin blood collection tubes (TERUMO, Shibuya, Tokyo, Japan) and mixed with fluorochrome-conjugated monoclonal antibodies against human cell surface antigens. To lyse and fix erythrocytes, FACS Lysing Solution (BD Biosciences, San Jose, CA, USA) was used. Flow cytometry data were obtained with a FACSAriaIII instrument (BD

Biosciences, San Jose, CA, USA). We followed standard immunophenotyping protocols from the Human Immunology Project[38] and the ONE Study[39].

**Differential expression analysis**. The identification of transcripts or proteins that were differentially expressed between patients with RA and HCs was conducted based on the empirical Bayes method using the limma R package. Age was used as the covariate in the linear model. For the transcripts, the RIN value was also included in the model. The false discovery rate was controlled based on $q$ values that were estimated with the $q$ value R package. We set the criterion for statistical significance at a $p$ value <0.05 and $q$ value <0.05. To test the expression data from immune cell subsets, surrogate variables estimated from transcriptional data from each subset were utilized to represent potential confounders because the RNA quality metric was not available for all samples. The single surrogate variable was estimated using the SVA method[40] with default parameters and used as a covariate in a linear model.

The RA probabilities calculated at each time point were fitted to the time after initiation of drug treatment via the cubic B-spline basis, and individuals were treated as a random effect using the duplicate Correlation function in limma[41]. The significance of the coefficients of the B-spline basis was tested by ANOVA.

**Development of RA diagnostic models**. We first regressed out the age effect from the gene expression, protein abundance, and cell abundance matrices because the age of the subject was potentially confounded by the disease label, as observed in Supplementary Table 1. The age effects were estimated based on the data from treatment-naive individuals of 45 patients with RA, 30 patients with primary Sjögren's syndrome (pSS), and 35 HCs, after removing batch effects for the expression and protein data. To increase confidence in the estimates of age effects, 30 pSS samples profiled with the same experimental batch with 30 patients with RA and 30 HCs[42] were included in the analysis. Each variable was fitted using the linear model with age and biological covariates, including gender and disease label. The estimated age and gender effects were regressed out from the data matrices. In the case of gene expression data, we also removed the effect of the RIN value from the data.

Based on an inspiration from the data-splitting procedure previously proposed[43], data were split into five subsets; four subsets were used for training, and the remaining subset was used as test data to estimate the accuracy of the model. By changing the groups used for testing and training, five models with potentially different parameters were generated. To avoid bias from the initial data split, we repeated this process three times. Thus, in total, 15 models were obtained for each data set. A partial least-squares regression (PLSR) was utilized to construct a diagnostic model, which allowed us to handle a large number of variables without prior feature selection and to interpret the model based on existing biological knowledge such as gene ontologies and reference transcriptomes of purified immune cells. PLSR does not select particular genes for prediction, but it identifies lateral predictive factors embedded in the data. These lateral factors potentially reflect biological systems such as cell abundance. Then, we can evaluate the importance of each gene in the prediction by assessing its contribution to lateral factors. In the PLSR model, the number of lateral components used for the prediction is a pre-defined parameter that must be specified. We performed tenfold cross-validation within the training set and optimized the number of components (from 1 to 10) based on the kappa statistic. We used the mean RA probability derived from those produced from the 15 model ensembles as a final output. The caret R package was used for PLSR modeling.

The PLSR models that were trained with data sets from the unmedicated individuals were applied to the drug-treated cohort to enumerate the RA probability for each sample. Before applying the PLSR model, the data were normalized with respect to age, gender, and/or the RIN value using the same coefficients that were used for normalization of the data from unmedicated individuals.

**Calculation of the variable contribution to the prediction**. The mathematical formulation of the PLSR model can be described as

$$\mathbf{X} = \mathbf{TP}^\mathrm{T} + \varepsilon_\mathrm{X},$$
$$\mathbf{Y} = \mathbf{TB}^\mathrm{T} + \varepsilon_\mathrm{Y},$$

where $\mathbf{X}$ is an $n \times m$ matrix of the omics measurement, $\mathbf{Y}$ is an outcome vector of length $n$, $\mathbf{T}$ is an $n \times l$ matrix of orthogonal scores, $\mathbf{P}$ is an $m \times l$ matrix of loading, $\mathbf{B}$ is a loading vector of length $l$, $\varepsilon_\mathrm{X}$ and $\varepsilon_\mathrm{Y}$ are the error terms, $n$ is the number of individuals, $m$ is the number of variables in the omics measurement, and $l$ is the number of orthogonal components. To estimate the relative variable contribution to the prediction, we first estimated the contribution of each orthogonal component to the prediction. The predictive function of the diagnosis given orthogonal components of $k$th individual is defined as

$$f_s(\mathbf{k}) = \sum_{i=1}^{s} t_{ki} \times b_i,$$

where $t_{ki}$ is an orthogonal score in the $k$th row and $i$th column of matrix $\mathbf{T}$, $b_i$ is a loading score for the $i$th orthogonal component, and $s$ is the number of orthogonal

components used for prediction. The mean squared error of prediction (MSEP) using $s$ orthogonal components is then estimated as

$$\text{MSEP}_s = \frac{1}{n}\sum_{k=1}^{n}(f_s(k) - y_k)^2$$

where $y_k$ is a diagnosis for the $k$th individual and $n$ is the number of individuals. The relative contribution of the $s$th orthogonal component to the prediction is estimated as

$$w_s = \frac{\text{MSEP}_s - \text{MSEP}_{s-1}}{\sum_{i=1}^{l}(\text{MSEP}_i - \text{MSEP}_{i-1})},$$

where $l$ is the number of orthogonal components in the model and $\text{MSEP}_0$ corresponds to MSEP of the model including only the intercept. Finally, the contribution of the $i$th variable to the prediction, $g_i$, is estimated as a weighted average of its loadings, and then $g_i$ is normalized to determine the relative contribution, $r_i$, as follows,

$$g_i = \frac{1}{l}\sum_{j=1}^{l} p_{ij} \times w_j,$$
$$r_i = \frac{g_i}{\sum_{j=1}^{m} g_j},$$

where $p_{is}$ is a loading in the $i$th row and $s$th column of matrix $\mathbf{P}$. We further normalized $r$ by scaling the maximum $r$ as 100. After calculating $r$ for each model in ensembles, we took the average of $r$ for each variable across models and applied it for the evaluation of influential variables. The procedure of MSEP estimation is described in ref. [44] and is implemented in the caret R package.

**Gene set overlap analysis.** The significance of the overlap between two gene sets was assessed with Fisher's exact test. For enrichment analysis with the MSigDB gene set collection[18], the Enrichment Map[45] was used for visualization of the results.

**Defining the residual molecular signature.** We first extracted variables with levels that were significantly different from the levels in HCs at the beginning of drug administration in responders to MTX, IFX, and TCZ ($p$ value <0.05 and $q$ value <0.05). The significant variables that were differentially expressed in the same direction compared with HC in all three treatment groups were defined as consensus RA signatures. Then, the consensus RA signatures were re-evaluated at week 24 after drug treatment for each treatment arm. The variables that continued to deviate ($p$ value <0.05 and $q$ value <0.05) in the same direction from the levels in HCs were defined as residual molecular signatures.

**Quantification of meta-expression features.** Meta-features for residual molecular signatures or pathway-level expression were estimated by the ssGSEA method[13] using the R GSVA package[46] with default parameters.

**Preparation of proteome reference.** The imputed intensity of label-free quantification for proteomes in 26 immune cells was obtained from the publication[14]. NCBI gene ID was assigned to each protein based on gene symbol and we kept proteins with NCBI gene ID for subsequent analysis. If the same gene ID was assigned to multiple proteins, the protein with the highest average intensity across immune cells was used. As a result of the filtering, the proteome reference contains 9379 proteins for 26 immune cells.

**Estimation of cell count contributions to RMSs.** The levels of residual molecular signatures quantified by the ssGSEA method[13] were then fitted to the 26 immunophenotypes using a multivariate linear regression model. Because the abundance of subsets of immune cells was highly correlated with each other, the coefficients of a linear model were potentially unidentifiable. To handle the collinearity problem, we first performed feature selection of model variables using the elastic net. The regularization parameter ($\lambda$) and mixing parameter ($\alpha$) that showed the best RMSE (root mean-squared error) in three repeated tenfold cross-validations was used. The actual parameters were as follows: $\alpha = 0.4$ and $\lambda = 0.01$ for protein residual molecular signatures (RMSs) downregulated in RA, $\alpha = 0$ and $\lambda = 0.03$ for protein RMSs upregulated in RA, $\alpha = 0$ and $\lambda = 0.01$ for transcriptional RMSs downregulated in RA, and $\alpha = 0$ and $\lambda = 0.03$ for transcriptional RMSs upregulated in RA. To assess the goodness of fit of the elastic net model, the sample labels of the transcriptome data were permuted 1000 times and fed into the same workflow, which included the step for hyperparameter tuning. Then, RMSEs from the permuted data were used as the null distribution to enumerate the $p$ value for the goodness of fit of the model. To estimate the variance of residual molecular signatures explained by the immunophenotypes, we re-fit a multivariate linear regression model with the immunophenotypes with none-zero coefficients in the elastic net model. The contribution of each cell type to residual molecular signatures was evaluated by averaging the sequential sums of squares over all orderings of regressors[47].

**Analysis of transcriptional RMS in other disease conditions.** Transcriptional RMSs were imported into NextBio (http://www.nextbio.com/) as of September 2016. Then, transcriptional RMS was compared with the public transcriptome studies conducted with Affymetrix GeneChip Human HG-U133 Plus 2.0 using the Running Fisher algorithm[48]. We retrieved comparative results for the public transcriptomes with the tag of "gene blood fraction" and "disease vs. normal". Next, we further manually removed studies related to any genetic modifications such as mutations, those with any compound treatments, those using purified immune cells or cultured immune cells, and those for non-disease conditions such as altitude change. Finally, the studies with a sample size less than ten were filtered out, resulting in the remaining 100 studies (Supplementary Fig. 19). The comparative statistics ($p$ values) for the studies with the same disease tagged by the NextBio database were merged using Stouffer's $z$-score method (Supplementary Data 4). The diseases for which the $z$-score fell into the upper and lower 2.5% quantiles of the overall distribution were designated as candidates for further investigation.

**Code availability.** R codes for RA diagnostic models have also been deposited in Synapse repository under accession code syn8483403.

**Data availability.** Accession codes: mRNA microarray, protein array, and immunophenotyping data have been deposited in the Gene Expression Omnibus (GEO) Data Bank under accession code ID GSE93777 and Synapse repository under accession code syn8483403.

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

## Acknowledgements

We thank Ms. Yoshiko Yogiashi, Ms. Yuki Otomo, and Ms. Harumi Kondo for assisting with the experiments.

## Author contributions

Conceptualization: S.T., K.S., Yo.K., T.M., A.Y., H.T. and T.T. Data curation: Y.O. Formal analysis: S.T., T.A., Y.N. and Y.O. Funding acquisition: A.Y. and T.T. Investigation: K.S., Yo.K., Mai.T. and R.K. Methodology: S.T., K.S., Yo.K., Mai.T., A.M., Ya.K., T.A., Y.N., Y.O., Mas.T., R.K., R.M., A.Y. and T.T. Project administration: K.S., A.Y. and T.T. Resources: K.S., Mas.T., A.M., Ya.K., Ke.Y., H.Y., Ku.Y., R.M., A.Y. and T.T. Supervision: K.S., T.M., A.Y., H.T. and T.T. Visualization: S.T. Writing—original draft preparation: S.T. Writing—review and editing: S.T., K.S., Mas.T., A.M., Ya.K., T.A., Y.N., T.M., Ke.Y., H.Y., Ku.Y., R.M., A.Y., H.T. and T.T.

## Additional information

**Competing interests:** S.T., Y.N., T.M. and H.T. were employed by Takeda Pharmaceutical Company Limited. Yo.K., T.A., Y.O., Mai.T. and R.K. are employed by Takeda Pharmaceutical Company Limited. Y.N. is employed by ONO Pharmaceutical. T. M. is employed by Nektar Therapeutics. H.T. is employed by FRONTEO. K.S. has received research grants from Eisai, Bristol-Myers Squibb, Kissei Pharmaceutical, and Daiichi Sankyo, and speaking fees from Abbie Japan, Astellas Pharma, Bristol-Myers Squibb, Chugai Pharmaceutical, Eisai, Fuji Film Limited, Janssen Pharmaceutical, Kissei Pharmaceutical, Mitsubishi Tanabe Pharmaceutical, Pfizer Japan, Shionogi, Takeda Pharmaceutical, and UCB Japan, consulting fees from Abbie, and Pfizer Japan. A.Y. has received speaking fees from Chugai Pharmaceutical, Mitsubishi Tanabe Pharmaceutical, Pfizer Japan, Ono Pharmaceutical, Maruho, and Novartis, and consulting fees from GSK Japan. Ku.Y. has received consultant fees from Pfizer, Chugai Pharma, Mitsubishi Tanabe Pharma, Abbvie, received honoraria from Pfizer, Chugai Pharma, Mitsubishi Tanabe Pharma, Bristol-Myers Squibb, Takeda Industrial Pharma, GlaxoSmithkline, Nippon Shinyaku, Eli lilly, Janssen Pharma, Eisai Pharma, Astellas Pharma, Actelion Pharmaceuticals and received research grants from Chugai Pharma, Mitsubishi Tanabe Pharma., and Glaxo Smith Kline. H.Y. has received research grants from Daiichi Sankyo, Takeda Pharmaceutical, Eisai, and Japan Blood Products Organization, and speaking fees from Abbie Japan, Bristol-Myers Squibb, Takeda Pharmaceutical, Chugai Pharmaceutical, and Eisai. T.T. has received research grants from Astellas Pharma Inc, Bristol-Myers K.K., Chugai Pharmaceutical Co. Ltd., Daiichi Sankyo Co. Ltd., Takeda Pharmaceutical Co. Ltd., Teijin Pharma Ltd., AbbVie GK, Asahikasei Pharma Corp., Mitsubishi Tanabe Pharma Co., Pfizer Japan Inc., and Taisho Toyama Pharmaceutical Co. Ltd., Eisai Co. Ltd., AYUMI Pharmaceutical Corporation, and Nipponkayaku Co. Ltd, and speaking fees from AbbVie GK., Bristol-Myers K.K., Chugai Pharmaceutical Co. Ltd., Mitsubishi Tanabe Pharma Co., Pfizer Japan Inc., and Astellas Pharma Inc., and Diaichi Sankyo Co. Ltd., and consultant fees from Astra Zeneca K.K., Eli Lilly Japan K.K., Novartis Pharma K.K., Mitsubishi Tanabe Pharma Co., Abbivie GK, Nipponkayaku Co. Ltd, Janssen Pharmaceutical K.K., Astellas Pharma Inc., and Taiho Pharmaceutical Co. Ltd. The remaining authors declare no competing interests.

