## [Peer Review File · Nature Communications]

Reviewers' comments:

Reviewer #1 (Remarks to the Author):

This draft presents a longitudinal study (2 years long) on a large cohort of RA patients, in an attempt to illuminate mechanisms of clinical remission.

In this Big Data study, the authors take a multi-omics perspective, collecting data from the transcriptome, proteome and immunophenotypes from patients, to build predictive models for distinguishing between RA patients and healthy controls.

The methods used are solid, and well-explained. I suggest addressing the question of disparity in the transcriptome and the proteome among the patients in this large cohort. For example, what is the gene expression correlation in the RA patients? Also, were technical replicates used in all/most samples? If not, why?

This study offers an influential way of merging clinical, biological and statistical analysis; hopefully more influential studies like this will be implemented.

One question to the authors is how the scientific community can use this large dataset/predictive modeling they built/performed? Could this dataset be available as an online searchable source? I suggest considering to make this available (including the statistical component) through an RShiny app.

Reviewer #2 (Remarks to the Author):

This review focuses on the statistical methodology used in this work, its appropriateness and its reproducibility.

The methodology looks globally correct but is somewhat confusing at places. Some details look also inconsistent throughout the text (e.g. cohort definition) or are simply missing (e.g. the elastic net parameters) and, as such, would prevent this work from being reproducible.

The detailed comments are included in an attached file.

Most of them require some clarifications from the authors to better support the validity of their work.

Reviewer #3 (Remarks to the Author):

In the manuscript entitled "Multi-omics monitoring of drug response in rheumatoid arthritis: in pursuit of molecular remission" the authors investigated patients with rheumatoid arthritis (RA) before and after initiation of methotrexate, infliximab or tocilizumab therapy by transcriptome, proteome and cell type analysis from peripheral blood samples and compared the data to healthy controls (HC) and for transcriptome data with other disease conditions. The authors conclude that molecular response data may provide more information for stable response prediction and that difference between remission and healthy condition displays treatment resistant signatures, which may implicate a therapeutic potential for RA.

The manuscript presents extensive data analysis, however, there are major issues, concerning

data analysis:

1. Comparison between RA and HC blood cell counts indicates elevated white blood cells, which is explained by elevated neutrophils and a relative decrease of all other cell types in RA blood. Increased cell counts of neutrophils in RA patients are frequently observed and explained by increased production in the bone marrow. This so called 'left shift' was recently demonstrated also for monocytes in RA (PMID:29191820). It can be expected that the changes in whole blood transcriptomes are predominantly reflecting such a shift of differential cell type composition in RA. Therefore, instead of comparing the differentially expressed genes with lists of functional annotations, it would be more informative to investigate how many of these genes are differentially expressed between neutrophils and other cell types of the peripheral blood.

Performing such a comparison would demonstrate that indeed almost all top candidate transcripts increased in RA are much higher expressed in neutrophils compared to lymphocytes and that almost all top transcripts decreased in RA are also lower or lowest in neutrophils and/or belong to other cell types. Annotation databases do not distinguish between gene expression in specific unstimulated cell types and functional patterns of stimulation. This information can only be extracted from the analysis of defined cell types and stimulation conditions. On this basis, the differentially expressed transcripts are no more supporting an interferon pathway trigger. If this trigger really would exist in RA blood transcriptomes, mapping the differentially expressed genes to transcriptomes of cells before and after IFN-stimulation should provide a more robust data interpretation and discussion. Data to perform such analyses may be in the hands of the authors (cell type transcriptomes) or can be retrieved from public repositories for Affymetrix HG-U133 Plus 2.0 arrays.

It would be helpful to distinguish more precisely between the up and the down regulated transcripts when characterizing their functional aspects and not only refer to all like "The genes highly contributed to the transcript-based model...". It is not clear from reading to what type of regulation the tRNA biosynthesis function belongs. It is necessary to look up possible candidates in the gene list of supplementary table 2, and even there it is necessary to look at which genes are up and which are down and which group is dominant. The detection of tRNA biosynthesis and mRNA splicing processes are probably mentioned because of reduced expression in RA. At least, when looking up in detail, it should be recognized that there is a reduced expression of riboproteins in neutrophils compared to lymphocytes and that these are reduced in RA along with the increase of neutrophils. This aspect should be presented as this patterning is very characteristic for neutrophils. Whether the "complement pathway" annotation is still a relevant functional association, has to be tested by focusing on the precise gene list of the used pathway annotation and comparing it with the differentially expressed transcripts between RA and HC whole blood and the differential patterning between different leukocyte cell types. That complement, which is part of the acute phase reaction produced in the liver, may also contribute but on the level of proteome data, can be expected. However, for the transcriptome results it should be sorted out whether this pathway is really functionally activated in blood cells.

2. In supplementary table 1 the individual parameters of the DAS28 (tender and swollen joint count, as well as VAS of patient self assessment) should be added like ESR is included. This will characterize more precisely the clinical situation (see also later). It would also help to better understand the remission index in figure 3 if the clinical parameters of TJC, SJC, VAS and ESR before and after treatment are displayed for each the 52 patients individually and integrated into the red/blue heatmap of figure 3a.

3. It is not astonishing that blocking the acute phase response by inhibiting the IL-6 pathway presents with more or less normal levels of the acute phase parameters. It would be more relevant to investigate the parameters that reflect the inflammation in the joint (for example tender and swollen joint count). These should be presented in supplementary table 4. Where there any parameters of inflammation in the joint and destruction of cartilage and bone in the proteome

screen included. What information can be extracted from these data? Are responses between IFX and TCZ still different? Including acute phase response dependent parameters like ESR into the assessment of response will produce a bias towards "polishing" the acute phase outcome but may not really improve joint inflammation much more than other biologics which target other factors that contribute to inflammation in the joint.

4. Considering the bias of the DAS28 parameters for assessment of disease activity and response in therapies that target IL-6 signaling, CR and MR should be critically reviewed in the paragraph of "MR defines long-term disease activities". Do individual parameters of the DAS28 (TJC, SJC, VAS, ESR) reflect the early CR and the long-term CR all in the same way or are the influences of ESR and CrP dominant for the interpretation of CR and MR relationship as it is currently suggested?

5. The patient numbers contributing to individual subgroups that categorize response outcome are very small for the individual treatment groups (for example TCZ with 3 patients that are in remission according to all three molecular assessments and only one is without MR despite of CR). The authors should refer to this problem and argue more carefully with respect to the lack of sufficient statistical power for their statements. Future studies with extended numbers of patients may change the situation a bit, not completely but strong enough so that the currently favored top candidate parameters may change to some extent. This would substantially influence the development of biomarker kits for the improvement of outcome prediction and provokes criticism for statements like the one in the discussion "... clinical assessments can be replaced with objective molecular biomarkers". More appropriate wording is necessary. May be, it is also worth to point out that analyzing not that many patients but individual diseases with much more profound molecular screens can strengthen the gain of insight into RA disease characteristics.

6. The authors have focused next on untreatable molecular signatures. Why skipping the characteristics of response? Is there a reduction of neutrophil left shift? An increase of lymphocytes? Do cell activation patterns, which may exist compared to HC, disappear in responders? Such analysis should also concentrate on reference transcriptomes and not only on gene lists that belong to annotation terms. Do transcriptome patterns and cell type analysis correspond to each other? Are the functions of proteome response patterns associated with the functions of transcriptome response patterns?

7. If the response patterns are defined, the untreatable patterns should be investigated in a similar way by testing their patterns for overlap with reference transcriptomes of cell types and stimulation conditions. Gene lists are not enlightening and lists of REACTOMEs with different names but identified by more or less the same genes (proteasome subunits) are not either. In that way, figures like supplementary figure 4 are not helpful and even misleading. Are these untreatable patterns mostly related to cell type characteristics or particular stimulation conditions? The splitting of gene lists according to GSEA defined functions and subsequent mapping of the gene sub-groups to transcriptome data is also misleading. Why not testing all untreatable genes in an unbiased way? If tested across different cell types from healthy controls, a preliminary rescreen that we performed, suggests that the majority of the upregulated genes are related to neutrophils. Whether these patterns may also reflect a trigger or function suggested by GSEA lists remains to be tested by analyzing reference transcriptomes of stimulation conditions. Otherwise, the gene lists of GSEA and functional annotations mostly reflect functional gene set entities which are preformed functional units of specialized cell types transcribed already under normal (unstimulated) condition but ready to react (increase, decrease, modulate) upon stimulatory triggers.

8. The cell type specific transcriptomes used to produce supplementary figure 8 are better applied if RA specific i) disease, ii) response and iii) untreatable transcriptome patterns are mapped to these reference cell types in order to test the cellular origin of these untreatable expression patterns.

9. What is the exact definition of the UTS genes? Only the 800 genes or all in supplementary table 5? If these genes are tested for cell type specific expression, most of the transcripts increased in RA belong to neutrophils and those decreased in RA to lymphocytes, suggesting that the dominant effect is related to a common inflammatory shift in blood cell count, which would be expected in many diseases.

In summary, the clinical study material and molecular raw data generation is excellent. The molecular analysis of transcriptome data is misleading when addressing functional interpretation. It is necessary to compare to reference transcriptomes instead of annotation databases as these generate confusing and misleading interpretations as described above. A preliminary reanalysis, which we performed, suggests that the main effect of differential expression in RA as well as in the untreatable transcript signature is related to the increase of neutrophils in RA (main inflammatory changes) with relative decrease of lymphocytes. Whether there are any additional effects (stimulation, left shift) has to be tested with appropriate reference transcriptomes that investigate these effects (for example PMID:29191820 or PMID:27570220). These transcriptome data should then be compared to the cell count analysis to confirm the results of both types of analysis. Sorting out the different transcriptional effects related to shifts in cellular composition or cell type specific changes related to stimulation effects may help to improve the screening for similarities in other diseases.

Minor points:

The language should be reviewed by a native speaker.

Supplementary Table 2c should contain a brief explanation for the abbreviations. For example what means "DC.rWBC"?

Page-5, 3rd line:

The term 'To understand biological molecules ...', it sounds better if it will be changed to 'To understand the function of biomolecules'

Page-7_legend of figure-1: The term 'red bar' is misleading because it's a vertical thin line
Page-10, 2nd paragraph headline: Print out the abbreviation MR.

Response to reviewers: NCOMMS-17-27832

We appreciate the reviewers' comments; they have been addressed in detail as outlined below. We have completed a re-analysis with reference transcriptome/proteome data that extend our understanding of the cellular origins of molecular alterations in RA with drug treatments and improve the clarity and rigor of data analysis.

For clarity, the reviewers' comments have been italicized.

Reviewers' Comments & Responses:

Reviewer #1 (Remarks to the Author):

This draft presents a longitudinal study (2 years long) on a large cohort of RA patients, in an attempt to illuminate mechanisms of clinical remission.

In this Big Data study, the authors take a multi-omics perspective, collecting data from the transcriptome, proteome and immunophenotypes from patients, to build predictive models for distinguishing between RA patients and healthy controls.

The methods used are solid, and well-explained. I suggest addressing the question of disparity in the transcriptome and the proteome among the patients in this large cohort. For example, what is the gene expression correlation in the RA patients? Also, were technical replicates used in all/most samples? If not, why?

Response:

We appreciate the reviewer's kind comments on our work. We agree with the reviewer that understanding the heterogeneity/disparity of molecular status in patients with RA is a very important topic and of substantial interest. Indeed, we have been investigating this and there were no clear sub-types that have distinct global molecular profile in our unmediated RA cohort. Because heterogeneity analysis would substantially impact the implementation of precision therapy for RA, we will further pursue to identify the sub-types and investigate their biological and clinical implications.

We have carefully considered the allocation of our limited budget to measurements of biological replicates versus technical replicates. Based on our previous experience (Sekiguchi et al., 2008) and the high productivity of the Affymetrix HG-U133 Plus 2.0 array (https://assets.thermofisher.com/TFS-Assets/LSG/brochures/hgu133_p2_technote.pdf), we decided to prioritize increasing the number of biological replicates and include technical replicates for a subset of samples. Essentially, if an experimental protocol is well designed, differentially expressed genes/proteins between cases and controls can be detected if biological variations are sufficiently larger than technical variations. The fact that we could identify a number of variables significantly associated with RA or drug treatments indicates that the biological variations in our samples were larger than the technical variations. To draw this conclusion, we carefully designed our experimental protocols with sample randomization so that any technical variables would not confound with the disease diagnosis or drug treatment groups. In addition, to gauge technical variability, we included technical duplicates for 20 and 19 samples for transcriptome and proteome analyses, respectively. The pairs of technical replicates showed greater concordance than those of non-technical replicates (**Figure R1**), indicating the high reproducibility of our sample preparation and measurement procedures. Furthermore, we previously confirmed the findings from our proteomic experiment using two conventional

assays: a latex turbidimetric immunoassay and ELISA (Murota et al., 2015). The success of the validation experiment for individual proteins indicates that our omics experiments were well designed to detect biological signals by effectively controlling for technical variability.

Figure R1. Correlation between pairs of technical and non-technical replicates.

This study offers an influential way of merging clinical, biological and statistical analysis; hopefully more influential studies like this will be implemented.

One question to the authors is how the scientific community can use this large dataset/predictive modeling they built/performed? Could this dataset be available as an online searchable source? I suggest considering to make this available (including the statistical component) through an RShiny app.

Response:

We would highly encourage researchers to re-use our data and conduct any follow-up studies on top of our findings. For this purpose, we deposited our data into Gene Expression Omnibus and are also detailing instructive pages in the Synapse database

(<https://www.synapse.org/>), which includes not only raw data but also ready-to-use processed data and predictive models. Unfortunately, we cannot construct interactive web pages using the time and human/computational resources allotted for this project, but we will continue to pursue the best way to share our data with a broad scientific community.

Reviewer #2 (Remarks to the Author):

This review focuses on the statistical methodology used in this work, its appropriateness and its reproducibility.

The methodology looks globally correct but is somewhat confusing at places. Some details look also inconsistent throughout the text (e.g. cohort definition) or are simply missing (e.g. the elastic net parameters) and, as such, would prevent this work from being reproducible.

The detailed comments are included in an attached file.

Most of them require some clarifications from the authors to better support the validity of their work.

The detailed comments below are listed essentially following the order of presentation in the text. Most of them require some clarifications from the authors to better support the validity of their work.

- *Untreated cohort The so-called unmedicated cohort is described in a somewhat inconsistent way. Line 68 and supplementary table 1 refer to 45 RA and 35 HCs but*

line 397 mentions 68 RA and 42 HCs. Maybe the difference comes from some excluded patients (see line 399) but this should be clarified by stating in the methods section how many patients are left out, specifically from RA and HCs. Besides, Line 481 mentions 75 patients which again differs from 45+35.

Response:

Sixty-eight patients with RA and 42 HCs are the total number of individuals who were subjected to the omics experiments conducted in this study. Of these, 67 patients with RA and 35 HCs were used for whole blood transcriptome, serum proteome, and immunophenotyping analyses, and data from 45 patients with RA without any drug treatments and 35 HCs were used for training models, which are described in **Supplementary Table 1**. Out of 68 patients, 49 were treated with MTX, IFX, or TCZ, and the whole blood transcriptome, serum proteome, and immunophenotypes were measured at multiple time points. Of note, two patients belonged to both the inadequate responders to IFX and the responders to TCZ group, and one patient was an inadequate responder to both IFX and TCZ. The demographic background and disease statuses of the patients with drug treatments are described in **Supplementary Table 5**. Out of 68 RA patients and 42 HCs, 14 patients with RA and 16 HCs were used for the transcriptome analysis of the immune cell subsets described in **Supplementary Table 8**. We have revised the methods section to include this information, and patient information will also be available in **Supplementary Table 10**, the Gene Expression Omnibus and Synapse repositories. The description of 75 patients on line 481 represents the number of patients with uremia in the public transcriptome data from GSE37171.

We have revised the sentence to clearly indicate 75 for the number of patients with uremia.

- *Treated cohort* The treated cohort (n=245) is mentioned in lines 249, 302, 497, 498 but is never actually described (accrual procedure, demographics, possible confounding factors, ...). Besides, some key results (figure 2, supplementary figure 6, ...) are reported on a small subset (claimed to be n=10 for each drug). Firstly, it is not clear why this analysis has been restricted to only 10 patients for each drug if 245 patients were under study. Secondly, it is not clear how the specific subsets of 10 patients have been chosen. Why were some patients left out? How were selected the 10 patients (x 2, R versus IR) for each drug? Thirdly, Supplementary Table 4 actually reports even fewer patients n=8 (IFX(IR)) and n=3 (TCZ(IR)). Why?

Response:

We reviewed the number of patients carefully and found that we misdescribed the total number of samples used for analyzing drug response. We left out 6 samples due to the potential inconsistency of sample sources with their clinical records and finally decided to use 239 samples. We have revised the total number of samples to 239. The demographics and background information for all patients with RA in the drug-treated cohort is described in **Supplementary Table 5**. Because we profiled whole blood samples at multiple time points (4 or 5 time points for responders and 2 or 3 time points for inadequate responders), the sample size of the drug response cohort was 239 in total. Based on the EULAR response criteria, we enrolled 10 responders who met our criteria for each drug. However, for inadequate responders, because most of the patients

responded well to IFX or TCZ in our cohort, only 8 or 3 patients met the criteria of inadequate responders in patients treated with IFX and TCZ, respectively.

- *Line 72 A clear motivation of the use of PLSR would be interesting to include. The problems at hand are standard binary classification problems (RA versus HC in the untreated cohort, R versus IR in the treated cohort) for which hundreds of methods exist (logistic regression, Bayes classifier, SVM, Random Forests, ... just to name a few). Picking one such method would be fine if the authors would at least state why they believe it is a relevant one. The "motivation" (line 486) with a single reference looks a bit short. In particular, PLSR is generally viewed as a dimensionality reduction technique for a regression problem (= continuous response). Here the authors address classification problems, even restricted to a binary response (e.g. R versus IR), without any specific focus on dimensionality reduction (at least through PLSR, see also below).*

Response:

We used PLSR in this study based on two characteristics that we considered important. The first point is to handle a large number of variables without prior feature selections since there are many arbitrary ways to select the features used for prediction, and thus, the inclusion of feature selection steps increases the hyper-parameters that need to be tuned. The second point is that the model should be easily interpreted based on existing biological knowledge, such as gene ontologies and reference transcriptomes of purified immune cells. The first point can be addressed by regularization-based methods, such as Elastic Net, and also by PLSR, which utilizes latent factors for prediction. Importantly,

PLSR can be applied to not only regression but also to classification, which is called partial least squares discriminant analysis. Although Elastic Net potentially shows good performance, we had a concern with Elastic Net regarding the second point. Elastic Net essentially aims to estimate a sparse model, potentially making the interpretation of the model difficult. Specifically, for instance, gene expression levels of particular gene groups are known to be highly correlated with each other, called co-expression modules. These modules represent biological systems and are enriched with various known biological pathways, and capturing such modules is thus critical for interpreting data. However, from the characteristic of Elastic Net, Elastic Net would give weight to a fewer number of informative genes within a module and no weights to other genes even if those genes are highly correlated with the informative genes. Thus, we expected that capturing biological systems that are co-fluctuated together via an Elastic Net model might be not straightforward. Conversely, PLSR does not select particular genes for prediction but rather finds lateral predictive factors embedded in the data. These lateral factors potentially reflect biological systems, such as cell abundance. Then, we can evaluate the importance of each gene to the prediction via assessing the gene's contribution to lateral factors (see details below). This procedure essentially finds all genes correlated with lateral factors and thus does not have the Elastic Net problem. As we expected, we could identify biological pathways and cell types significantly associated with variables highly contributing to the prediction in PLSR models. The sparseness of Elastic Net would benefit the actual implementation of prediction models in the clinic because we can make predictions based on a fewer number of genes. However, in this study, we prioritized interpreting biological mechanisms using the model and therefore chose RLSR.

- *Line 81 The use of an ensemble of 15 models to get a more robust prediction looks sound but the reported results with such an ensemble are presented in a somewhat misleading fashion. In particular, the results reported in Figure 1 C and Supplementary Figure 1 are **no longer test** accuracies since the whole dataset (apparently 35 HCs + 45 RA) has been used (through a 3x5-CV protocol) in order to compute the average prediction. In other words, results reported in Figure 1 C and Supplementary Figure 1 are **training set** accuracies, as far as the ensemble is concerned, and should be reported as such. An additional collection of new and independent samples/patients would be needed to assess the classification performance of such an ensemble on a real test set not seen before. Fortunately, since these models are eventually used to compute RA odds on the treated cohort (which is the central topic here) this is not critical for the rest of the manuscript. Yet, the captions of Figure 1 C and Supplementary Figure 1 + the related text should be fixed.*

Response:

We agree with the reviewer that **Figure 1C** represents the training set accuracies and clarified this in the legend. **Supplementary Figure 1** indicates the average of cross-validation accuracies, which is essentially the same as that in **Figure 1B**. We clarified this in the legend of **Supplementary Figure 1**. In addition, we revisited the estimation of null distributions and found that the cross-validation accuracies of models trained with permuted data should also be averaged in each ensemble, as we did for an ensemble of 15 models trained with actual data. The null distributions estimated via the revised scheme show smaller variance, and the significant performances of the actual models became more evident (**Figure R2**). We replaced **Supplementary Figure 1** with **Figure R2**.

Figure R2. Null distributions of averaged cross-validation accuracies. The red bars indicate the averaged cross-validation accuracies of 15 model ensembles with non-permuted data.

As we described above, we built the model based on data from 45 patients with RA who had not received any medications, while patients who had been treated with any medications were left out of the training process. With this respect, we examined whether the models introduce upward bias on RA probability for the samples used in the training process. To do this, we compared the RA odds between 45 patients who were used in the training and 22 patients who had been treated with medications but didn't respond to them and found that there were no significant differences in the RA odds estimated from any of the models (**Figure R3**). These results indicate that our ensemble models are reasonably accurate and not significantly biased in the training samples. We have added **Figure R3** in the revised manuscript as **Supplementary Figure 6**.

Figure R3. Comparison of RA odds between training and test samples. P-values from Welch's t-test are indicated above the boxplots.

- *Line 118 + Supplementary figure 2 The authors should describe in the Methods section how they actually computed the contribution of each specific covariate to the ensemble model prediction. A precise mathematical formula would be even better.*

Response:

We have added a detailed procedure for computing the contribution of each specific covariate to the prediction in the methods section as follows:

The mathematical formulation of the PLSR model can be described as,

$$X = TP^T + \varepsilon_X,$$

$$Y = TB^T + \varepsilon_Y,$$

where X is the $n \times m$ matrix of omics measurement, Y is the outcome vector of length n , T is the $n \times l$ matrix of orthogonal scores, P is the $m \times l$ matrix of loading, B is the loading vector of length l , ε_X and ε_Y are the error terms, n is the number of individuals, m is the number variables in omics measurement, and l is the number of orthogonal components. To estimate the relative variable contribution to the prediction, we first estimated the contribution of each orthogonal component to the prediction. The predictive function of diagnosis given the orthogonal components of the k th individual is defined as

$$f_s(k) = \sum_{i=1}^s t_{ki} \times b_i,$$

where t_{ki} is the orthogonal score in the k th row and i th column of matrix T , b_i is the loading score for the i th orthogonal component, and s is the number of orthogonal components used for prediction. Then the mean squared error of prediction (MSEP) using s orthogonal components was estimated as

$$MSEP_s = \frac{1}{n} \sum_{k=1}^n (f_s(k) - y_k)^2,$$

where y_k is the diagnosis for the k th individual, and n is the number of individuals. The relative contribution of the s th orthogonal component to the prediction was estimated as

$$w_s = \frac{MSEP_s - MSEP_{s-1}}{\sum_{i=1}^l (MSEP_i - MSEP_{i-1})},$$

where l is the number of orthogonal components in the model, and $MSEP_0$ corresponds to the MSEP of the model including only intercept. Finally, the contribution of the i th variable to the prediction, g_i , was estimated as the weighted average of its loadings, and g_i was then normalized to obtain the relative contribution, r_i , as follows,

$$g_i = \frac{1}{l} \sum_{j=1}^l p_{ij} \times w_j,$$

$$r_i = \frac{g_i}{\sum_{j=1}^m g_j},$$

where p_{is} is the loading in the i th row and s th column of matrix P . We further normalized

r by scaling the maximum r to 100. After calculating r for each model in the ensembles, we averaged the r values for each variable across models and used this for the evaluation of influential variables. The procedure for MSE_P estimation is described in (Mevik and Cederkvist, 2004) and is implemented in the caret R package.

- *Figure 2 C This interesting figure reports differences (in RA odds) globally for the treated cohort (and 3 specific drugs). Since this is a multi-omics approach, 3 sub-figures are proposed respectively for WB transcript, serum protein and cell count with somewhat consistent trends. It would be interesting to analyze whether the 3 models (one for each data source) are actually consistent on a per sample (= per patient) basis rather than only globally. To which extent is the same patient correctly predicted to be R versus IR from the 3 models? If not so, why?*

Response:

We have conducted pair-wise comparisons of the treatment effects on RA odds estimated by the 3 models. We found a significant consistency between the protein-based model and the cell count-based model, while the transcript-based model showed modest positive correlations with the others (**Figure R4**). Additionally, the responders and inadequate responders were well separated via the protein-based model and the cell count-based model (**Figure R5**). Conversely, although the responders tended to show more reductions in RA odds in the transcript-based model than the inadequate responders, the difference did not reach the statistical significance level. This was not because drug treatments effectively normalize the transcriptional signatures associated with RA in both responders and inadequate responders but rather that the treatment effects on the transcriptional signatures are limited even in responders. Indeed, we identified a sizable number of genes

that remain to be differentially expressed even after drug treatments in the responders, suggesting that molecular unmet needs mainly exist in transcriptomes. We have added **Figure R4** and **Figure R5** in the main figures as **Supplementary Figure 11** and **Figure 2d**, respectively.

Figure R4. Correlation between the treatment effects on RA odds based on three molecular classes.

Figure R5. Correlation between the model-based assessments of drug effects and the clinical definition of drug response.

- *Line 205 "By using stringent criteria (see Methods), we found 800 transcripts"..." Referring to line 526, a q-value < 0.25 does not look stringent at all. Why not considering a more standard threshold: q-value < 0.05? This would also be consistent with FDR < 0.05, a common threshold the authors themselves use (e.g. lines 122, 172, ...)*

Response:

We have revised the analyses with the use of q-value < 0.05 as a criterion of significance. The numbers of untreatable variables in each treatment arm were slightly decreased, but the untreatable variables shared among the three treatment arms remained the same.

- *Supplementary Figure 4 I wonder **what** can actually be inferred from such a "messy cloud" and the authors themselves do not say a word about it (apart from referring to this figure in line 241 without any comment). Which kind of negative control is considered here (I believe, none)? How is this figure informative? So, I suggest to plainly skip this figure or to argue much more convincingly about it. If kept, I would recommend to produce it with FDR < 0.05.*

Response:

We agree with the reviewer's comment. We removed **Supplementary Figure 4** from the revised manuscript.

- *Line 302 and line 560 Which are the actual λ and α values used? How were they chosen?*

Response:

We used the λ and α values that showed the lowest root-mean-square error (RMSE) in three repeated 10-fold cross-validation procedures. The actual parameters are as follows: $\alpha= 0.4$ and $\lambda= 0.01$ for protein residual molecular signatures (RMSs) down-regulated in RA, $\alpha= 0$ and $\lambda= 0.03$ for protein RMSs up-regulated in RA, $\alpha = 0$ and $\lambda= 0.01$ for transcriptional RMSs down-regulated in RA, and $\alpha = 0$ and $\lambda= 0.03$ for transcriptional RMSs up-regulated in RA. We have described the λ and α values in the method section.

- *Line 542 What does it mean exactly to be concordant? How was this actually computed?*

Response:

The aim of this analysis was to identify untreatable genes that are common to all three drugs. Therefore, we intended to describe that we ensured that untreatable genes were differentially expressed in the same direction compared to the HCs in all three treatment groups. We have clarified the statement to indicate that we ensured the directional concordance of differentially expressed genes in the three treatment groups. In fact, none of the genes were filtered with this criterion.

- *Line 561, Line 567 "We constrained the coefficients of the variables to be positive." "... with the positive coefficients in the elastic net model." A multivariate linear regression model penalized with elastic net has absolutely no guarantee to be restricted to positive coefficients. Besides, there is generally no need for such a positivity constraint. For instance, if one would like to interpret the model coefficient associated to a particular covariate as a measure of variable importance, the absolute value of the model coefficient (which can be either positive or negative) is very commonly used. So, the authors should clarify **why** they wanted to enforce a positivity constraint on the model coefficients and **how** they did it. It would also be interesting to mention the specific solver they used (or at least a reference to a package implementing such a solver) to fit a multivariate regression with Elastic Net.*

Response:

We conducted Elastic Net using the R glmnet package developed by researchers at Stanford University. This implementation allows users to set upper and lower bounds of coefficients (https://web.stanford.edu/~hastie/glmnet/glmnet_alpha.html). We believed that it is a natural assumption that the gene expression levels in whole blood are the summation of the gene expression levels in each immune cell in whole blood. Therefore, to follow this assumption, we placed a constraint on the coefficients in the model so that immune cells can only positively contribute to the transcriptional levels in whole blood. Based on the reviewer's comment, we reconsidered this scenario and agreed with the reviewer's thought that this constraint is not necessary. Specifically, the cell type with a negative coefficient can be interpreted as that expressing lower levels of the transcriptional signature compared to other cell types. We rebuilt the Elastic Net model without any constraints on coefficients and found that the result concurs with the reference

transcriptome data for each cell type (**Figure R22**). We appreciate the reviewer's comment that led us to better understanding our data.

Reviewer #3 (Remarks to the Author):

In the manuscript entitled "Multi-omics monitoring of drug response in rheumatoid arthritis: in pursuit of molecular remission" the authors investigated patients with rheumatoid arthritis (RA) before and after initiation of methotrexate, infliximab or tocilizumab therapy by transcriptome, proteome and cell type analysis from peripheral blood samples and compared the data to healthy controls (HC) and for transcriptome data with other disease conditions. The authors conclude that molecular response data may provide more information for stable response prediction and that difference between remission and healthy condition displays treatment resistant signatures, which may implicate a therapeutic potential for RA.

The manuscript presents extensive data analysis, however, there are major issues, concerning data analysis:

1. Comparison between RA and HC blood cell counts indicates elevated white blood cells, which is explained by elevated neutrophils and a relative decrease of all other cell types in RA blood. Increased cell counts of neutrophils in RA patients are frequently observed and explained by increased production in the bone marrow. This so called 'left shift' was recently demonstrated also for monocytes in RA (PMID:29191820). It can be expected that the changes in whole blood transcriptomes are predominantly reflecting such a shift of differential cell type composition in RA. Therefore, instead of comparing the differentially expressed genes with lists of functional annotations, it would be more

informative to investigate how many of these genes are differentially expressed between neutrophils and other cell types of the peripheral blood.

Performing such a comparison would demonstrate that indeed almost all top candidate transcripts increased in RA are much higher expressed in neutrophils compared to lymphocytes and that almost all top transcripts decreased in RA are also lower or lowest in neutrophils and/or belong to other cell types. Annotation databases do not distinguish between gene expression in specific unstimulated cell types and functional patterns of stimulation. This information can only be extracted from the analysis of defined cell types and stimulation conditions. On this basis, the differentially expressed transcripts are no more supporting an interferon pathway trigger. If this trigger really would exist in RA blood transcriptomes, mapping the differentially expressed genes to transcriptomes of cells before and after IFN-stimulation should provide a more robust data interpretation and discussion. Data to perform such analyses may be in the hands of the authors (cell type transcriptomes) or can be retrieved from public repositories for Affymetrix HG-U133 Plus 2.0 arrays.

It would be helpful to distinguish more precisely between the up and the down regulated transcripts when characterizing their functional aspects and not only refer to all like “The genes highly contributed to the transcript-based model...”. It is not clear from reading to what type of regulation the tRNA biosynthesis function belongs. It is necessary to look up possible candidates in the gene list of supplementary table 2, and even there it is necessary to look at which genes are up and which are down and which group is dominant. The detection of tRNA biosynthesis and mRNA splicing processes are probably mentioned because of reduced expression in RA. At least, when looking up in detail, it should be recognized that there is a reduced expression of riboproteins in

neutrophils compared to lymphocytes and that these are reduced in RA along with the increase of neutrophils. This aspect should be presented as this patterning is very characteristic for neutrophils. Whether the “complement pathway” annotation is still a relevant functional association, has to be tested by focusing on the precise gene list of the used pathway annotation and comparing it with the differentially expressed transcripts between RA and HC whole blood and the differential patterning between different leukocyte cell types.

That complement, which is part of the acute phase reaction produced in the liver, may also contribute but on the level of proteome data, can be expected. However, for the transcriptome results it should be sorted out whether this pathway is really functionally activated in blood cells.

Response:

We agree with the reviewer that transcriptional changes likely reflect alterations in the cellular composition of whole blood. To clarify the contributions of each cell type to the transcriptional changes observed in patients with RA, we assessed the expression profiles of up- or down-regulated genes based on our own reference transcriptomes of 14 purified immune cells measured via the Affymetrix HG-U133 Plus 2.0 array (**Figure R6**) and the public reference proteomes of 26 immune cells (Rieckmann et al., 2017). Specifically, the single-sample GSEA method (Barbie et al., 2009) was used to merge the gene expression levels of multiple genes into single meta-expression. This meta-expression strategy is particularly useful for understanding the expression profiles of a small number of genes, such as genes in pathways across immune cells, whereas a standard enrichment analysis has very low statistical power for a small number of genes. Using this meta-expression approach, we evaluated the expression profiles of up- or down-regulated genes that are

important for discriminating RA vs HC in the PLSR model and pathways enriched with those genes. As the reviewer expected, the genes up-regulated in RA are highly expressed in neutrophils, and the genes down-regulated in RA are lowly expressed in neutrophils (**Figure R7**). Furthermore, pathways enriched with transcriptional changes in RA, including interferon signatures, tRNA biosynthesis, and mRNA splicing, show higher expression in neutrophils compared to that in other cell types (**Figure R8**). These results were also confirmed at the protein levels based on a public proteome reference (**Figure R9**). These results suggest that the increase in neutrophil counts is the major reason underlying the transcriptional changes in both gene and pathway levels. The model based on cell counts in this study is based on absolute white blood counts and the relative abundance of each cell type in white blood. Since white blood is a mixture of cell types that have distinct expression profiles and functions, as the reviewer suggested, absolute white blood counts would lead to the misinterpretation of the results. Therefore, to obtain a more interpretable result, we decomposed the white blood counts into absolute counts of each cell type and rebuilt the model based on absolute counts and the relative composition of cell types. The updated model indicates that the increase in both the absolute count and relative composition of neutrophils is the major factor for discriminating RA vs HC (**Figure R10**). As the results from the cell count-based model and transcript-based model both point to the elevation of neutrophils, we described the 'left shift' as the biological interpretation of diagnostic models in the main text.

Figure R6. Hierarchical clustering of gene expression profiles from 14 immune cells.

Figure R7. Expression patterns of up and down-regulated RA signatures in immune cell subsets.

Figure R8. Expression patterns of pathways enriched with up- and down-regulated RA genes in immune cell subsets.

Figure R9. Protein expression profiles of important transcripts across 26 immune cells.

Meta-expression features for the key transcripts up-regulated or down-regulated in RA were calculated separately using the ssGSEA method based on the protein expression profiles of 26 immune cells and standardized across immune cells.

Figure R10. The top ten important cell types for discriminating between patients with RA and HC. The red and blue vertical bars indicate variables that were upregulated and downregulated in RA, respectively. The error bars represent variabilities in the contribution to the model prediction that originated from the model ensemble.

2. In supplementary table 1 the individual parameters of the DAS28 (tender and swollen joint count, as well as VAS of patient self assessment) should be added like ESR is included. This will characterize more precisely the clinical situation (see also later). It would also help to better understand the remission index in figure 3 if the clinical parameters of TJC, SJC, VAS and ESR before and after treatment are displayed for each the 52 patients individually and integrated into the red/blue heatmap of figure 3a.

Response:

We have revised **Supplementary Table 1 (Table R1)** and **Figure 3A (Figure R11)** to include the individual parameters of DAS28.

	HC	RA	SjS
n	35	45	30
AGE (mean (sd))	41.11 (9.82)	56.33 (15.20)	61.07 (10.80)
GENDER = M (%)	5 (14.3)	7 (15.6)	1 (3.3)
RF = positive (%)		29 (64.4)	
ACPA = positive (%)		29 (64.4)	
CRP [mg/dl] (mean (sd))		1.31 (1.68)	
ESR [mm/hr] (mean (sd))		54.87 (32.33)	
DAS28 CRP (mean (sd))		4.22 (1.21)	
DAS28 ESR (mean (sd))		5.14 (1.22)	
SDAI (mean (sd))		21.77 (14.13)	
CDAI (mean (sd))		20.46 (13.31)	
HAQ-DI (mean (sd))		0.92 (0.70)	
TJC28 (mean (sd))		5.93 (5.52)	
SJC28 (mean (sd))		5.84 (5.79)	
Physician GA, VAS [mm] (mean (sd))		36.42 (21.65)	
Subject GA, VAS [mm] (mean (sd))		50.36 (23.25)	

Table R1. Updated Table S1 with inclusion of the individual parameters of DAS28

Figure R11. Updated Figure 3A with integration of the individual parameters of DAS28 CDAI, and HAQ.

3. It is not astonishing that blocking the acute phase response by inhibiting the IL-6 pathway presents with more or less normal levels of the acute phase parameters. It would be more relevant to investigate the parameters that reflect the inflammation in the joint (for example tender and swollen joint count). These should be presented in supplementary table 4. Where there any parameters of inflammation in the joint and destruction of cartilage and bone in the proteome screen included. What information can

*be extracted from these data? Are responses between IFX and TCZ still different?
Including acute phase response dependent parameters like ESR into the assessment of
response will produce a bias towards “polishing” the acute phase outcome but may not
really improve joint inflammation much more than other biologics which target other
factors that contribute to inflammation in the joint.*

Response:

We have revised **Supplementary Table 4 (Table R2)** to include the individual parameters of the DAS28. We agree with the reviewer that assessments without the inclusion of acute phase response-dependent parameters are better for further understanding the drug response. To enable a more detailed characterization of drug response, in addition to DAS28 (ESR), we used CDAI, which defines clinical remission without considering acute phase response-dependent parameters, and HAQ-DI, which defines functional remission. These indices are included in **Figure 3A (Figure R11)** and are also focused upon in the remaining analysis in **Figure 3**. The revised analysis indicates that TCZ is more effective for inducing the clinical remission defined by DAS28 (ESR) but not as effective for CDAI and HAQ-DI, while patients treated with IFX archived remission in mostly DAS28 (ESR), CDAI and HAQ-DI (**Figure R11**). We have revised the analysis for relating molecular remission and clinical and functional remission using these three indices.

	IFX(IR)	IFX(R)	MTX(IR)	MTX(R)	TCZ(IR)	TCZ(R)
n	8	10	11	10	3	10
AGE (mean (sd))	50.75 (18.88)	51.10 (16.91)	66.91 (8.34)	58.70 (10.63)	68.67 (11.72)	58.00 (14.89)
GENDER = M (%)	1 (12.5)	0 (0.0)	3 (27.3)	2 (20.0)	1 (33.3)	0 (0.0)
RF = positive (%)	6 (85.7)	6 (75.0)	5 (45.5)	7 (70.0)	2 (100.0)	7 (87.5)
ACPA = positive (%)	6 (85.7)	6 (75.0)	4 (36.4)	7 (70.0)	2 (100.0)	6 (75.0)
CRP [mg/dl] (mean (sd))	1.33 (1.03)	1.49 (1.05)	2.01 (2.51)	0.99 (1.04)	4.19 (2.92)	1.70 (1.92)
ESR [mm/hr] (mean (sd))	48.25 (34.87)	60.10 (31.24)	56.00 (38.31)	50.30 (32.40)	103.00 (50.86)	67.80 (34.05)
DAS28 CRP (mean (sd))	4.06 (1.18)	5.46 (1.21)	4.24 (0.97)	4.15 (0.75)	5.80 (2.60)	4.86 (1.38)
DAS28 ESR (mean (sd))	4.81 (1.34)	6.35 (1.25)	5.04 (0.96)	4.97 (0.76)	6.75 (2.65)	5.88 (1.25)
SDAI (mean (sd))	19.71 (10.96)	35.50 (18.90)	20.84 (11.29)	18.96 (7.47)	45.46 (30.77)	27.07 (17.80)
CDAI (mean (sd))	18.38 (10.83)	34.01 (18.06)	18.84 (9.49)	17.97 (7.42)	41.27 (27.86)	25.37 (16.76)
HAQ-DI (mean (sd))	0.72 (0.46)	1.50 (0.43)	1.09 (0.77)	0.85 (0.61)	1.92 (1.04)	1.49 (0.88)
TJC28 (mean (sd))	4.62 (4.31)	11.40 (7.81)	4.09 (3.30)	5.20 (3.46)	13.33 (11.55)	7.80 (6.55)
SJC28 (mean (sd))	4.75 (4.37)	11.30 (8.00)	5.27 (5.61)	5.00 (3.30)	14.00 (12.12)	7.70 (6.46)
PAIN.VAS (mean (sd))	52.25 (17.56)	67.56 (19.60)	59.73 (20.60)	41.10 (25.24)	76.00 (9.17)	64.20 (28.08)
Physician GA, VAS [mm] (mean (sd))	37.62 (18.15)	48.30 (22.17)	34.00 (23.82)	29.40 (11.50)	63.33 (41.74)	37.30 (24.63)
Subject GA, VAS [mm] (mean (sd))	52.38 (15.26)	64.80 (18.84)	60.73 (21.23)	48.30 (15.53)	76.00 (9.17)	61.40 (27.73)

Table R2. Updated Table S4 with the inclusion of individual DAS28 parameters

4. *Considering the bias of the DAS28 parameters for assessment of disease activity and response in therapies that target IL-6 signaling, CR and MR should be critically reviewed in the paragraph of “MR defines long-term disease activities”. Do individual parameters of the DAS28 (TJC, SJC, VAS, ESR) reflect the early CR and the long-term CR all in the same way or are the influences of ESR and CrP dominant for the interpretation of CR and MR relationship as it is currently suggested?*

Response:

To clarify the relationships between the molecular and clinical remission indices, we computed the correlation between the molecular remission in each molecular class and the clinical remission indices at 24 weeks of treatment. Molecular remissions defined based on serum proteins are strongly correlated with DAS28 (ESR) but not with CDAI and HAQ-DI. Cell count- and transcript-based remissions were not associated with any

clinical remission indices, suggesting that these measures reflect the characteristics of patients, which is not included in the clinical indices. Then, we further examined the individual parameters that drive an association between protein-based remission and DAS28 (ESR), as the reviewer suggested. We found that ESR was significantly associated with molecular remission based on proteins (**Figure R12**). We note that although the associations did not reach the significance level, all parameters were closer to the normal state in patients with MR than those with non-MR. We also examined the individual parameters of DAS28 that reflect the relationships between long-term CR and MR. The result indicates that ESR and TJC28 are associated with the MR status (**Figure R13**), suggesting that MR influences not only ESR but also the long-term inflammation status in joints.

Figure R12. Relation between molecular remission and individual parameters of DAS28 and CDAI at 24 weeks. The dashed line represents a p-value corresponding to 0.05.

Figure R13. Relation between the number of classes archived for molecular remission and individual parameters of DAS28 and CDAI in biologics-treated patients at 90 weeks in the follow-up. The dashed line represents a p-value corresponding to 0.05.

5. The patient numbers contributing to individual subgroups that categorize response outcome are very small for the individual treatment groups (for example TCZ with 3 patients that are in remission according to all three molecular assessments and only one is without MR despite of CR). The authors should refer to this problem and argue more carefully with respect to the lack of sufficient statistical power for their statements. Future studies with extended numbers of patients may change the situation a bit, not completely but strong enough so that the currently favored top candidate parameters may change to some extent. This would substantially influence the development of biomarker kits for the improvement of outcome prediction and provokes criticism for statements like the one in the discussion "... clinical assessments can be replaced with objective molecular

biomarkers". More appropriate wording is necessary. May be, it is also worth to point out that analyzing not that many patients but individual diseases with much more profound molecular screens can strengthen the gain of insight into RA disease characteristics.

Response:

We agree with the reviewer's comment that our statement is too deterministic for the result based on the limited samples. We described a limitation of the molecular signatures as prediction makers and placed more emphasis on the deep molecular understanding of RA pathogenesis in the context of drug response in the discussion.

6. The authors have focused next on untreatable molecular signatures. Why skipping the characteristics of response? Is there a reduction of neutrophil left shift? An increase of lymphocytes? Do cell activation patterns, which may exist compared to HC, disappear in responders? Such analysis should also concentrate on reference transcriptomes and not only on gene lists that belong to annotation terms. Do transcriptome patterns and cell type analysis correspond to each other? Are the functions of proteome response patterns associated with the functions of transcriptome response patterns?

Response:

In the response to the reviewer's comment, we characterized the effects of drug treatments on the levels of each transcript, protein, and cell type (**Figure R14**). Approximately 600 transcripts were differentially expressed in patients treated with IFX or TCZ, but no genes exceed the significance criteria for MTX treatment (**Figure R14**). Transcriptional changes induced by IFX and TCZ treatments occur mainly in genes that

are highly or lowly expressed in neutrophils (**Figure R15**), suggesting that the neutrophil signature is normalized by drug treatments. The decrease in neutrophil abundance was confirmed with actual cell count data (**Figure R16**), indicating that the drug treatments reduced the left shift in neutrophils observed in un-medicated patients with RA. TCZ treatment had a strong effect on serum proteins (**Figure R15**), which was also indicated by the protein model-based analysis. MTX affects a greater number of proteins than IFX, but a sizable number of those proteins are altered in a direction away from the healthy state (**Figure R15**). This directional inconsistency corresponds to the moderate reduction in RA odds in MTX-treated patients assessed by the protein-based model. Pathway analysis of serum proteins showed that proteins involved in complement pathways are enriched in proteins affected by IFX and TCZ but not by MTX (**Figure R17**). Complement pathways are also enriched in proteins associated with un-medicated patients with RA, suggesting that IFX and TCZ specifically target pathways aberrantly activated in RA. These results are included in **Figure 2** and **Supplementary Figure 12** in the revised manuscript.

Figure R14. Number of variables affected by drug treatments (24 weeks vs 0 weeks).

Figure R15. Meta-expression of transcripts affected by IFX or TCZ in purified immune cells.

Figure R16. Drug effects on the abundance of neutrophils and NK cells. The asterisks represent a p-value <0.05 and FDR <0.05.

Figure R17. Pathway analysis of serum proteins affected by drug treatments. The asterisks represent a p-value <0.05 and FDR <0.05.

7. If the response patterns are defined, the untreatable patterns should be investigated in a similar way by testing their patterns for overlap with reference transcriptomes of cell types and stimulation conditions. Gene lists are not enlightening and lists of REACTOMEs with different names but identified by more or less the same genes (proteasome subunits) are not either. In that way, figures like supplementary figure 4 are not helpful and even misleading. Are these untreatable patterns mostly related to cell type characteristics or particular stimulation conditions? The splitting of gene lists according to GSEA defined functions and subsequent mapping of the gene sub-groups to transcriptome data is also misleading. Why not testing all untreatable genes in an unbiased way? If tested across different cell types from healthy controls, a preliminary rescreen that we performed, suggests that the majority of the upregulated genes are related to neutrophils. Whether these patterns may also reflect a trigger or function suggested by GSEA lists remains to be tested by analyzing reference transcriptomes of stimulation conditions. Otherwise, the gene lists of GSEA and functional annotations mostly reflect functional gene set entities which are preformed functional units of specialized cell types transcribed already under normal (unstimulated) condition but ready to react (increase, decrease, modulate) upon stimulatory triggers.

Response:

We are thankful for the reviewer's insightful suggestions. We re-analyzed untreatable molecular signatures to elucidate associated clinical phenotypes, functions and cell types in an unbiased manner. First, we found that the levels of untreatable molecular signatures approach those of healthy controls even though they are still significantly different (**Figure**

R18). From this, we believe that residual molecular signature (RMS) is a more appropriate term for these molecular signatures than the untreatable molecular signature. In terms of clinical phenotypes associated with RMS, we found that transcriptional RMS and protein RMS showed weak correlations with DAS28-ESR and CDAI at week 0 (**Figure R19**). However, these trends were not observed at week 24. Conversely, morning stiffness is associated with transcriptional RMS at week 24, but this association is not present at week 0 (**Figure R20**). Together, these results indicate that relations between RMSs and specific clinical phenotypes are not supported robustly in our data. Therefore, we revised the clinical implication of RMSs to not focus on morning stiffness or other phenotypes but rather to emphasize that RMSs are the molecular characteristics of patients with RA that are largely independent of disease severities and cannot be normalized completely with the current symptomatic treatments.

We next investigated cell types associated with RMS using reference transcriptomes and proteomes from purified immune cells. Transcriptional RMS up-regulated in RA is highly expressed in neutrophils and monocytes, and transcriptional RMS down-regulated in RA is lowly expressed in these cell types (**Figure 21**). In contrast, protein RMS does not show specific expression in some cell types. We then next examined whether cell counts can explain the expression levels of transcriptional and protein RMSs. Cell counts are estimated to explain 40% and 48% of the variation in up and down-regulated transcriptional RMSs and 33% and 37% of the variation in up and down-regulated protein RMSs, respectively. These fractions are significantly higher than random expectations (permutation p-value < 0.001). The abundance of neutrophils and monocytes largely explain the levels of transcriptional and protein RMSs (**Figure 22**). Considering both the expression specificity and associations with cell counts, the increases in neutrophil and monocyte counts would be the major driver for transcriptional RMS. Indeed, we found

significant correlations between transcriptional RMS and neutrophil or monocytes counts at week 0 (**Figure 23**). However, at week 24, correlations between transcriptional RMS and neutrophil counts were weaker than those at week 0, while the relationship with monocytes remained the same. This suggests that transcriptional RMS after drug treatment is not only due to the left shift in neutrophils but also to the left shift in monocytes.

We further asked whether cell composition variabilities explain the observed differences in the levels of RMSs between RA and HC. To evaluate this, we removed cell count effects from RMSs and contrasted the residuals between RA and HC. When cell count was accounted for, the variabilities in RMSs explained by the RA diagnosis were decreased but still significantly high for all three drugs (**Figure 24**). This raised the possibility that expression changes at the cellular level might also contribute to RMS. To test this, we compared the expression profiles of purified immune cells from RA and HC. Transcriptional RMSs tended to be differentially expressed in a manner concordant with whole blood in the varieties of cell subsets tested (**Figure 25**). Together, our re-analysis clarified the major contribution of cell composition and cellular level expression to the presence of transcriptional RMS in RA.

Figure R18. Levels of transcriptional and protein RMS before and after drug treatments.

Figure R19. Association with RMSs and disease severity indices before and after drug treatments. Asterisks represent a p-value of 0.05.

Figure R20. Association with transcriptional RMSs and various clinical phenotypes before and after drug treatments. Asterisks represent a p-value of 0.05.

Figure R21. Expression profiles of transcriptional RMSs in purified immune cells.

Figure R22. Variance in RMSs explained by cell counts.

Figure R23. Correlation between the RMSs and cell counts of key cell types before and after drug treatments. The dashed line represents a p-value of 0.05.

Figure R24. Contribution of cell counts to the differential expression of RMSs between RA and HC.

Figure R25. Differential expression profiles of transcriptional RMSs in a variety of purified immune cells from RA and HC.

8. The cell type specific transcriptomes used to produce supplementary figure 8 are better applied if RA specific i) disease, ii) response and iii) untreatable transcriptome patterns are mapped to these reference cell types in order to test the cellular origin of these untreatable expression patterns.

Response:

In the revised manuscript, we annotated transcripts and genes based on the reference transcriptomes and proteomes, as we have shown in this response.

9. What is the exact definition of the UTS genes? Only the 800 genes or all in supplementary table 5? If these genes are tested for cell type specific expression, most of the transcripts increased in RA belong to neutrophils and those decreased in RA to lymphocytes, suggesting that the dominant effect is related to a common inflammatory shift in blood cell count, which would be expected in many diseases.

Response:

The definition of RMS is genes/proteins/cell types whose levels are significantly different (FDR<0.05) between RA and HC both before (week 0) and after (week 24) treatment.

We then assessed the intersection of RMS for all three drugs, resulting in the

identification of 800 genes and 13 proteins as consensus RMSs, which were used for the subsequent analyses. The RMS for each drug is also listed in **Supplementary Table 7**. As we described above, we have added a detailed investigation to understand the cellular origins of RMS based on reference transcriptomes/proteomes, cell counts, and transcriptomes from purified immune cells.

In summary, the clinical study material and molecular raw data generation is excellent. The molecular analysis of transcriptome data is misleading when addressing functional interpretation. It is necessary to compare to reference transcriptomes instead of annotation databases as these generate confusing and misleading interpretations as described above. A preliminary reanalysis, which we performed, suggests that the main effect of differential expression in RA as well as in the untreatable transcript signature is related to the increase of neutrophils in RA (main inflammatory changes) with relative decrease of lymphocytes. Whether there are any additional effects (stimulation, left shift) has to be tested with appropriate reference transcriptomes that investigate these effects (for example PMID:29191820 or PMID:27570220). These transcriptome data should then be compared to the cell count analysis to confirm the results of both types of analysis. Sorting out the different transcriptional effects related to shifts in cellular composition or cell type specific changes related to stimulation effects may help to improve the screening for similarities in other diseases.

Response:

We appreciate the reviewer's constructive comments throughout the manuscript, which led to substantial improvements in the biological interpretation of our data.

Minor points:

The language should be reviewed by a native speaker.

Response:

We appreciate your suggestion. All text in the revised manuscript has been reviewed by experts.

Supplementary Table 2c should contain a brief explanation for the abbreviations. For example what means “DC.rWBC”?

Response:

Thank you for bringing this to our attention. “DC.rWBC” represents dendritic cell counts normalized by total white blood cell count. We have added a description for each cell variable.

Page-5, 3rd line:

The term ‘To understand biological molecules ...’, it sounds better if it will be changed to ‘To understand the function of biomolecules’

Page-7_ legend of figure-1: The term ‘red bar’ is misleading because it’s a vertical thin line
Page-10, 2nd paragraph headline: Print out the abbreviation MR.

Response:

We have revised the corresponding text as the reviewer suggested. We appreciate the reviewer's careful reading of our manuscript.

Reference

- Barbie, D.A., Tamayo, P., Boehm, J.S., Kim, S.Y., Moody, S.E., Dunn, I.F., Schinzel, A.C., Sandy, P., Meylan, E., Scholl, C., et al. (2009). Systematic RNA interference reveals that oncogenic KRAS-driven cancers require TBK1. *Nature* 462, 108–112.
- Mevik, B.-H., and Cederkvist, H.R. (2004). Mean squared error of prediction (MSEP) estimates for principal component regression (PCR) and partial least squares regression (PLSR). *J. Chemom.* 18, 422–429.
- Murota, A., Suzuki, K., Kassai, Y., Miyazaki, T., Morita, R., Kondo, Y., Takeshita, M., Niki, Y., Yoshimura, A., and Takeuchi, T. (2015). Serum proteomic analysis identifies interleukin 16 as a biomarker for clinical response during early treatment of rheumatoid arthritis. *Cytokine* 78, 87–93.
- Rieckmann, J.C., Geiger, R., Hornburg, D., Wolf, T., Kveler, K., Jarrossay, D., Sallusto, F., Shen-Orr, S.S., Lanzavecchia, A., Mann, M., et al. (2017). Social network architecture of human immune cells unveiled by quantitative proteomics. *Nat. Immunol.* 18, 583–593.
- Sekiguchi, N., Kawauchi, S., Furuya, T., Inaba, N., Matsuda, K., Ando, S., Ogasawara, M., Aburatani, H., Kameda, H., Amano, K., et al. (2008). Messenger ribonucleic acid expression profile in peripheral blood cells from RA patients following treatment with an anti-TNF-alpha monoclonal antibody, infliximab. *Rheumatology (Oxford)*. 47, 780–788.

REVIEWERS' COMMENTS:

Reviewer #1 (Remarks to the Author):

The authors have addressed all comments satisfactorily.

Reviewer #2 (Remarks to the Author):

All my suggestions and remarks about the previous version of this manuscript have been carefully addressed.

Reviewer #3 (Remarks to the Author):

From my point of view, the authors have dealt precisely and purposefully with the outstanding questions and suggestions.

Due to the improvements and changes made, I therefore see no need for further changes of the manuscript and would suggest acceptance for publication.

REVIEWERS' COMMENTS:

Reviewer #1 (Remarks to the Author):

The authors have addressed all comments satisfactorily.

Reviewer #2 (Remarks to the Author):

All my suggestions and remarks about the previous version of this manuscript have been carefully addressed.

Reviewer #3 (Remarks to the Author):

From my point of view, the authors have dealt precisely and purposefully with the outstanding questions and suggestions. Due to the improvements and changes made, I therefore see no need for further changes of the manuscript and would suggest acceptance for publication.

We would greatly appreciate insightful and constructive reviews.